# Learning Latent Variable Models via Jarzynski-adjusted Langevin Algorithm

**James Cuin**
Department of Mathematics
Imperial College London
London, UK
jamie.cuin23@imperial.ac.uk

**Davide Carbone**
Laboratoire de Physique de l'Ecole Normale Supérieure,
Université PSL, CNRS,
Sorbonne Université, Université de Paris
Paris, France
davide.carbone@phys.ens.fr

**O. Deniz Akyildiz**
Department of Mathematics
Imperial College London
London, UK
deniz.akyildiz@imperial.ac.uk

## Abstract

We utilise a sampler originating from nonequilibrium statistical mechanics, termed here Jarzynski-adjusted Langevin algorithm (JALA), to build statistical estimation methods in latent variable models. We achieve this by leveraging Jarzynski's equality and developing algorithms based on a weighted version of the unadjusted Langevin algorithm (ULA) with recursively updated weights. Adapting this for latent variable models, we develop a sequential Monte Carlo (SMC) method that provides the maximum marginal likelihood estimate of the parameters, termed JALA-EM. Under suitable regularity assumptions on the marginal likelihood, we provide a nonasymptotic analysis of the JALA-EM scheme implemented with stochastic gradient descent and show that it provably converges to the maximum marginal likelihood estimate. We demonstrate the performance of JALA-EM on a variety of latent variable models and show that it performs comparably to existing methods in terms of accuracy and computational efficiency. Importantly, the ability to recursively estimate marginal likelihoods—an uncommon feature among scalable methods—makes our approach particularly suited for model selection, which we validate through dedicated experiments.

## 1 Introduction

Real world data often contains latent structures (Whiteley et al., 2025) that can be described well with a latent variable model (LVM). As a result, LVMs have become ubiquitous in modern statistical research, from natural language processing (Wang et al., 2023) to bioinformatics (Shen et al., 2009), due to their flexibility in capturing complex and hidden processes underlying observed data. It is therefore of significant interest to *fit* LVMs, i.e., learn its parameters from data.

A particularly prominent estimation paradigm for LVMs is that of maximum marginal likelihood estimation (MMLE) (Dempster et al., 1977). Given some fixed observed data $y$, the task of MMLE is to compute the maximum likelihood estimate of the parameters, denoted $\theta$, by maximising the marginal likelihood $p_\theta(y)$. This intractable quantity is defined through marginalising the joint likelihood of the observed data $y$ and the latent variable $x$, over the latent variable $x$. More precisely, let $y \in \mathbb{R}^{d_y}$ be the observed data, $x \in \mathbb{R}^{d_x}$ be the latent variables, and $\theta \in \mathbb{R}^{d_\theta}$ be the parameters of interest. Given

39th Conference on Neural Information Processing Systems (NeurIPS 2025).

some fixed observed data $y$, we define the joint likelihood function $p_\theta(x, y) : \mathbb{R}^{d_x} \times \mathbb{R}^{d_\theta} \to \mathbb{R}$. Our main task in this paper is to develop methods for finding the maximisers of the *marginal* likelihood $p_\theta(y) : \mathbb{R}^{d_\theta} \to \mathbb{R}$, i.e., we aim to solve

$$\theta^\star \in \underset{\theta \in \mathbb{R}^{d_\theta}}{\arg\max} \log p_\theta(y), \quad \text{where} \quad p_\theta(y) = \int_{\mathbb{R}^{d_x}} p_\theta(x, y) \mathrm{d}x. \tag{1}$$

Indeed, the integral in (1) is often analytically intractable or numerically expensive, making direct maximisation of the marginal likelihood infeasible in many relevant applications.

Historically, the gold standard for solving the problem in (1) is the Expectation-Maximisation (EM) algorithm (Dempster et al., 1977), which is a two-step iterative procedure that alternates between estimating an expectation $Q(\theta, \theta_{k-1}) = \mathbb{E}_{p_{\theta_{k-1}}(x|y)}[\log p_\theta(x, y)]$ (E-step) and updating the parameter via $\theta_k \mapsto \arg\max_\theta Q(\theta, \theta_{k-1})$ (M-step). This algorithm requires the implementation of two general procedures, integration for the E-step and maximisation for the M-step, which are generally intractable for complex statistical models. This has resulted in many numerical strategies for implementing these steps efficiently, which gave rise to a large number of variants and approximations of the EM algorithm. Early examples include simulation based approaches for the E-step for models where it is possible to sample from the posterior $p_\theta(x|y)$ for a given $\theta$ (Wei and Tanner, 1990), termed Monte Carlo EM. Similarly, the M-step is often approximated using numerical optimisation techniques as exact maximisation is intractable (Liu and Rubin, 1994; Meng and Rubin, 1993; Lange, 1995). In general, however, these schemes are still impossible to implement as the posterior is rarely amenable to exact sampling, which resulted in the use of Markov chain Monte Carlo (MCMC) algorithms for the E-step. There has been a significant body of work in this direction, see, e.g., Atchadé et al. (2017); Caffo et al. (2005); Delyon et al. (1999); Fort and Moulines (2003); De Bortoli et al. (2021); Gruffaz et al. (2024). Most notably, De Bortoli et al. (2021) used unadjusted Langevin chains for the E-step and gradient descent schemes for the M-step, which is most related to our work. This approach may result in long running times for the Markov chain approximating the E-step, which also complicates the analysis. To overcome such limitations, a significant body of work developed an interacting particle systems approach to solve the MMLE problem (Kuntz et al., 2023; Caprio et al., 2025; Sharrock et al., 2024; Akyildiz et al., 2025) including extensions for nondifferentiable models (Encinar et al., 2025) and accelerated schemes (Lim et al., 2024; Oliva and Akyildiz, 2024) with applications to generative modelling (Wang et al., 2025; Oliva et al., 2025; Marks et al., 2025) and inverse problems (Glyn-Davies et al., 2025). These works use a *particle system* instead of a MCMC chain for the E-step, which is computationally efficient. However, most of these methods run the particles in the space of latent variables independently (except Sharrock et al. (2024)) - which could also be improved with further interaction. Moreover, these methods do not allow an easy computation of the model likelihood $p_\theta(y)$ for a given $\theta$.

There are alternative, closer to our approach, sequential Monte Carlo (SMC)-based methods. Johansen et al. (2008) considers an SMC algorithm on an extended target measure that concentrates on the MMLE solution. Crucinio (2025) provides a general SMC method that bears similarities to ours, but is designed for a specific parameter update mechanism (rather than general optimisers). In the context of particle filtering, Hamiltonian Monte Carlo approaches have been explored, as in Septier and Peters (2015).

**Contributions.** To address the issues mentioned above, in this paper, we build on a numerical technique to sample probability paths, which we term Jarzynski adjusted Langevin algorithm (JALA). JALA is a Langevin Monte Carlo (LMC) method to sample from time-varying probability measures, corrected using a SMC rather than Metropolis steps. The key idea is to run biased dynamics, specifically that of the unadjusted Langevin algorithm (ULA) with no Metropolis correction, and subsequently correct for the bias in sampling via an exponentially weighted factor referred to as a *Jarzynski factor*. Building on this idea:

- In Section 2, we formulate the Jarzynski adjusted Langevin algorithm (JALA) for sampling from time-varying sequence of distributions. The algorithm is based on the sampler developed in Carbone et al. (2023) and is closely related to sequential Monte Carlo (SMC) samplers (Del Moral et al., 2006) – and can be seen as a weighted-ensemble of the ULA.

- In Section 3, using the JALA as the core component, we propose a numerical method for EM, using JALA, which we term Jarzynski adjusted Langevin algorithm for EM (JALA-EM). This method uses the JALA for estimating the gradient of the marginal likelihood, which

is then used to update the parameters via a gradient-based optimiser. The resulting JALA-EM algorithm is a sequential Monte Carlo method that provides the maximum marginal likelihood estimate of the parameters.

- In Section 4, we provide a convergence analysis, under log-concavity and Polyak-Łojasiewicz (PŁ) conditions. In particular, we provide a nonasymptotic analysis of the JALA-EM method which is implemented via stochastic gradient descent (SGD). We utilise the convergence analysis of SGD algorithms to prove our nonasymptotic result. This is just a first step, as any other gradient-based optimiser can be used in the JALA-EM algorithm and their theoretical properties can be used to prove convergence of the JALA-EM algorithm.

- Finally, in Section 5, we demonstrate the performance of JALA-EM on a variety of LVMs and provide empirical evidence that it successfully estimates model parameters for various regression models and that it can also be used in model selection unlike the methods above.

**Computational Cost and Code.** Experiments were run on a personal computer and a Google Colab T4 GPU. The code can be found in https://github.com/jamescuin/jala-em.

## 2 Jarzynski-adjusted Langevin algorithm

We first introduce here the Jarzynski adjusted Langevin algorithm (JALA) which is a sampling method for time-varying probability distributions. The main idea behind the method is to run ULA on these time-varying potentials and correct the resulting bias (due to time-dependence) using the Jarzynski equality. The resulting JALA is a sampler for a sequence of evolving measures.

Consider a sequence of target distributions $(\pi_k)_{k \geq 0}$ on $\mathbb{R}^{d_x}$ and assume that we wish to use a ULA-based strategy to sample from these distributions. As opposed to the case with a static target measure, the problem is significantly harder, as a naive application of the ULA, i.e., iterating $X_{k+1} = X_k - h\nabla_x U_k(x) + \sqrt{2h}\xi_{k+1}$ where $U_k(x) = -\log \pi_k(x)$, will not yield samples from the target distribution $\pi_k$ at time $k$. In fact, even introducing a Metropolis step would not correct the bias (even asymptotically), as the distributions $(\pi_k)_{k \geq 0}$ are evolving. One possible solution is the idea of employing a Jarzynski-based correction for sampling from time-evolving measures which was studied by Carbone et al. (2023). As opposed to the setting therein, we first introduce the method for time-varying potentials $(U_k)_{k \geq 0}$, and then we will apply it to the case of LVMs in Section 3.

**Proposition 1** (Adapted from Proposition 1 of Carbone et al. (2023))**.** *Assume that $Z_k :=$ $\int e^{-U_k(x)}\mathrm{d}x < \infty$ for all $k \geq 0$. Let the sequences $(X_k)_{k \geq 0}$ and $(A_k)_{k \geq 0}$ be given by the iteration rule*

$$\begin{cases} X_{k+1} = X_k - h\nabla_x U_k(X_k) + \sqrt{2h}\,\xi_{k+1}, & X_0 \sim \pi_0, \\ A_{k+1} = A_k - \alpha_{k+1}(X_{k+1}, X_k) + \alpha_k(X_k, X_{k+1}), & A_0 = 0, \end{cases} \tag{2}$$

*where $\{\xi_k\}_{k \in \mathbb{N}_0}$ are i.i.d $\mathcal{N}(0_d, I_d)$, and we define*

$$\alpha_k(x_l, x_r) = U_k(x_l) + \frac{1}{2}(x_r - x_l) \cdot \nabla U_k(x_l) + \frac{h}{4}|\nabla U_k(x_l)|^2. \tag{3}$$

*Then, for all $k \in \mathbb{N}_0$ and a test function $\varphi : \mathbb{R}^{d_x} \to \mathbb{R}$, we have*

$$\mathbb{E}_{\pi_k}[\varphi(X)] = \frac{\mathbb{E}\left[\varphi(X)e^{A_k}\right]}{\mathbb{E}\left[e^{A_k}\right]}, \qquad Z_k = Z_0\mathbb{E}\left[e^{A_k}\right], \tag{4}$$

*where the expectations on the right-hand side, of both equations, are over the law of the joint process $(X_k, A_k)$, as defined through (2).*

See Appendix B.1 for a proof. Eq. (4) provides a powerful tool for estimating expectations with respect to the time-evolving measures $\pi_k$. In particular, it allows us to compute the expectation of any test function $\varphi$ with respect to the target distribution $\pi_k$ by running the joint process $(X_k, A_k)$ and using the Jarzynski factor $e^{A_k}$ to correct for the bias introduced by the unadjusted Langevin dynamics. In practice, the expectation in (4) cannot be computed, hence a particle algorithm can be used, which makes the method an instance of an SMC method. This result then can be used for estimating parameters of an LVM or fitting energy-based models (EBMs) as observed in Carbone et al. (2023), as time-varying potentials $(U_k)_{k \geq 0}$ can be indexed by some parameter $\theta_k$ and the sampler can be used to compute the gradient of the log-marginal likelihood.

---
**Algorithm 1** JALA-EM
---

1: **Inputs**: Observed data $y \in \mathbb{R}^{d_y}$; potential energy $U$; number of particles $N \in \mathbb{N}$; number of iterations $K \in \mathbb{N}$; step-size $h > 0$; gradient-based optimiser OPT; ESS threshold $C > 0$; and set of walkers $\left\{X_0^i\right\}_{i \in [N]}$ sampled from $p_{\theta_0}(x|y)$.

2: Initialise $A_0^i = 0$ for all $i \in [N]$.

3: **for** $k = 0, \dots, K-1$ **do**

4: $\quad w_k^i = e^{A_k^i} / \sum_{j=1}^N e^{A_k^j}, \quad$ for all $i \in [N]$. $\qquad\qquad$ *// compute weights*

5: $\quad g_k = \sum_{i=1}^N w_k^i \nabla_\theta U\left(\theta_k, X_k^i\right).$ $\qquad\qquad\qquad$ *// estimate the gradient*

6: $\quad \theta_{k+1} = \text{OPT}(\theta_k, g_k)$ $\qquad\qquad\qquad\qquad\qquad$ *// update the parameter*

7: $\quad$ **for** $i = 1, \dots, N$ **do**

$$X_{k+1}^i = X_k^i - h \nabla_x U\left(\theta_k, X_k^i\right) + \sqrt{2h}\xi_{k+1}^i,$$
$$A_{k+1}^i = A_k^i - \alpha_{k+1}\left(X_{k+1}^i, X_k^i\right) + \alpha_k\left(X_k^i, X_{k+1}^i\right),$$

$\quad$ where $\xi_k \sim \mathcal{N}(0, I_{d_x})$ and $\alpha_k$ is defined as in (2) with $U_k := U(\theta_k, \cdot)$.

8: $\quad$ **end for**

9: $\quad$ Resample w.r.t. $w_{k+1}^i$ and set $A_{k+1}^i = 0$ if $\text{ESS}_{k+1} < C$.

10: **end for**

11: **Outputs**: $\theta_K, \{w_K^i\}_{i=1}^N, \{X_K^i\}_{i=1}^N$

---

**Remark 1.** *The definition of $A_k$ update in relation to SMC can be interpreted as a particular case of Eq. (12) in Del Moral et al. (2006), or later on Sec. 2.4 in Heng et al. (2020), where the backward transition kernel $L_{t-1}(X_t, X_{t-1})$ is chosen to be $M_t(X_t, X_{t-1})$, given that $M_t(X_{t-1}, X_t)$ is the forward transition kernel (i.e. ULA kernel in the case of (2)). Exemplary illustrations of this situation in the context of MCMC design can be found in Nilmeier et al. (2011) and in Fig. 2 of Schönle et al. (2025). A theoretical motivation for this particular choice is that in the limit $h \to 0$ one recovers $A_{k+1} = A_k + \partial_t U_t(X_k)h + \mathcal{O}(h^{3/2})$, meaning that the continuous time limit for the evolution of the log-weights is coherent with Jarzynski work up to $\mathcal{O}(h^{1/2})$, i.e. the discretization error of Euler-Maruyama ULA, cfr. Carbone et al. (2024). Moreover, for sufficiently small $h$ the ULA kernel becomes reversible and "1-step" ergodic (cfr. Mattingly et al. (2002)), ensuring that for a finite sample size the estimator (4) is not affected by variance and bias issues due to poor exploration. In practice, different strategies for tuning $h$ are available, see for instance Kim et al. (2025).*

## 3 Jarzynski-adjusted Langevin algorithm for MMLE

We now outline Jarzynski adjusted Langevin algorithm for EM (JALA-EM), in which the key idea is to approximate the nonequilibrium corrections required by Jarzynski's identity through a population of particles that evolve under ULA, whilst simultaneously performing parameter updates.

Let $y$ be the observed data and $p_\theta(x, y)$ be a joint distribution of an LVM. For the ease of notation, let $U(\theta, x) = -\log p_\theta(x, y)$ where $U$ will be referred to as joint potential. Our aim is to compute the minimisers of $V(\theta) = -\log p_\theta(y)$ wrt. $\theta$ as $y$ is fixed. Given that this cannot be done analytically, an appealing scheme would be to implement a gradient based optimiser, using $\nabla_\theta \log p_\theta(y)$. However, as we have discussed, this gradient cannot be computed in closed form either. Nevertheless, for sufficiently regular models (see, e.g., Douc et al. (2014, Appendix D)), one can write this gradient as

$$\nabla_\theta V(\theta) = -\nabla \log p_\theta(y) = \int \nabla_\theta U(\theta, x) p_\theta(x|y) \mathrm{d}x = \mathbb{E}_{p_\theta(x|y)}\left[\nabla_\theta U(\theta, x)\right], \qquad (5)$$

where $U(\theta, x) = -\log p_\theta(x, y)$ and $p_\theta(x|y)$ is the posterior distribution of latent variables. This identity (which is known as the *Fisher's identity*) is behind many recent algorithms to implement MMLE procedures, see, e.g., De Bortoli et al. (2021) for an implementation of EM based on this identity (using ULA chains to sample from $p_\theta(x|y)$).

Instead of an MCMC-based method, we aim at using the JALA to approximate the expectation in (5). This can indeed be done straightforwardly using Proposition 1, by replacing $U_k(x)$ with $U(\theta_k, x)$ and letting $(\theta_k)_{k \geq 0}$ evolve with some discrete-time protocol (e.g., SGD). As noted in the last section, this

computation can only be performed using a particle scheme and then setting $\varphi(x) := \nabla_\theta U(\theta_k, x)$. This is the main idea behind JALA-EM. To develop our method, we instantiate the method using $N$ particles, resulting in the system:

$$X_{k+1}^i = X_k^i - h\nabla_x U\left(\theta_k, X_k^i\right) + \sqrt{2h}\xi_k^i, \tag{6}$$

$$A_{k+1}^i = A_k^i - \alpha_{k+1}\left(X_{k+1}^i, X_k^i\right) + \alpha_k\left(X_k^i, X_{k+1}^i\right), \tag{7}$$

where $\alpha_k$ is defined as in (3) with $U_k := U(\theta_k, \cdot)$. At iteration $k$, the gradient can be estimated by computing the normalised weights $w_k^i = e^{A_k^i}/\sum_{j=1}^N e^{A_k^j}$, and then computing the weighted average

$$g_k = \sum_{i=1}^N w_k^i \nabla_\theta U\left(\theta_k, X_k^i\right). \tag{8}$$

This can then be used for a generic first-order optimiser. We outline the method in Algorithm 1. A few remarks are in order for our method.

**Remark 2.** *We note that the method (6)–(7) is an* SMC *method, where $A_k$ are the unnormalised log-weights. It is thus vulnerable to standard problems of* SMC *samplers, such as weight degeneracy. For this to not happen, the sequence $(\theta_k)_{k\geq 0}$ needs to 'vary slowly' which can be achieved with small step-sizes or adaptive approaches, see, e.g.,* Zhou et al. (2016). *To keep track of the degeneracy, we use the effective sample size (*ESS*) as $ESS_k = 1/\sum_{i=1}^N (w_k^i)^2$* (Elvira et al., 2022). *This quantity takes values in between $1$ and $N$. Initially, since $A_0^i = 0$, we have $ESS_0 = N$, but it decreases with $k$, as generally observed in* SMC. *When this happens, a resampling operation is triggered (see Appendix C.1 for details).*

## 4 Nonasymptotic bounds for JALA-EM

In this section, we establish nonasymptotic error rates for our algorithm, where the optimiser OPT is assumed to be a first-order method, specifically, stochastic gradient descent (SGD):

$$\text{OPT}_{\gamma_k}(\theta_k, g_k) = \theta_k - \gamma_k g_k, \tag{9}$$

where $(\gamma_k)_{k\geq 0}$ is a sequence of step-sizes and $(g_k)_{k\geq 0}$ are the estimates of $\nabla V(\theta) := -\nabla_\theta \log p_\theta(y)$ for $(\theta_k)_{k\geq 0}$.

**Remark 3.** *Our framework would allow for deriving nonasymptotic bounds using any optimiser (notably adaptive optimisers), as soon as the analysis for biased gradients (e.g.* Surendran et al. (2024)*) is available. This is in contrast to similar methods like* Kuntz et al. (2023); Akyildiz et al. (2025) *where theory strictly requires gradient descent type update for the parameters, albeit in practice adaptive optimisers are used in experiments. See Appendix A.1 for a more thorough discussion of this point.*

### 4.1 Bounding the MSE of the stochastic gradient

In order to establish nonasymptotic rates for JALA-EM, the first challenge is to notice that our stochastic gradient estimate given by (8) is *biased*, as opposed to the usual SGD setting. Therefore, we need to first provide an analysis of our estimator w.r.t. the true gradient, which requires the following assumptions.

**A1.** *We assume that (i) the sequence $(A_k)_{k\geq 0}$ has finite exponential moments, i.e. $\sup_{k\geq 0} \mathbb{E}[e^{2A_k}] < \infty$, (ii) there exists a measurable set $A \subset \mathbb{R}^d$, independent of $k$, such that: $\text{Leb}(A) > 0$ (positive Lebesgue measure) and $\sup_\theta \sup_{x\in A} U(\theta, x) \leq M$ for some $M < \infty$, (iii) the gradient satisfies $\|\nabla_\theta U\|_\infty = \sup_{(\theta,x)\in\mathbb{R}^{d_\theta}\times\mathbb{R}^{d_x}} \|\nabla_\theta U(\theta, x)\|_\infty < \infty$.*

Three remarks are in order regarding A1. First, the finite exponential moments are about the weights of the SMC method, specifically, we require weights to be bounded, which is usually a standard assumption. Secondly, A1(ii) immediately implies $Z_k \equiv Z_{\theta_k} \geq \int_A e^{-U(\theta_k,x)}dx \geq \int_A e^{-M}dx = e^{-M}\text{Leb}(A) > 0$, and just correspond to ask $p_{\theta_k}(x|y)$ to not concentrate mass in a zero measure set. Thirdly, we require the gradient to be bounded, as it acts as a test function in our importance sampling (IS)-type estimator. This assumption can be dropped at the expense of significantly complicating the analysis, see, e.g., Agapiou et al. (2017). We thus keep this assumption for simplicity.

Assume that we run the system (2) with $U_k(\cdot) := U(\theta_k, \cdot)$. At iteration $k$, for any $\theta_k \in \mathbb{R}^{d_\theta}$, we have that

$$\nabla_\theta V(\theta_k) = \mathbb{E}_{p_{\theta_k}(x|y)} \left[ \nabla_\theta U(\theta_k, x) \right] = \frac{\mathbb{E}\left[ \nabla_\theta U(\theta_k, X_k) e^{A_k} \right]}{\mathbb{E}\left[ e^{A_k} \right]},$$

where in the last equality we have used Proposition 1 with $\varphi(x) = \nabla_\theta U(\theta_k, x)$ and so the expectations on the right-hand side are over the law of the joint process, as defined in (2). In practice, however, we run a particle system given in (6)–(7) and estimate this gradient as in (8). Our estimator is an IS-type estimator and we can prove the following proposition to bound the mean-squared error (MSE).

**Proposition 2.** *Under A1, for any sample size, $N \geq 1$, we have that*

$$\mathbb{E}\left[ \|\nabla V(\theta_k) - g_k\|^2 \right] \leq \frac{4C \|\nabla_\theta U\|_\infty^2}{N},\tag{10}$$

*for $k \geq 0$, where $C = \sup_k \mathbb{E}\left[ \left( e^{A_k} \right)^2 \right] / \left( \mathbb{E}\left[ e^{A_k} \right] \right)^2 = \sup_k C_k < \infty$.*

A self-contained proof can be found in Appendix B.2.

**Remark 4.** *We note that Proposition 2 is for our scheme without resampling. The full scheme with resampling requires nontrivial analysis which we leave for future work. In light of this, one has to choose $N = \mathcal{O}(1/\varepsilon)$ to obtain that $\mathrm{MSE} \leq \varepsilon$. With resampling, the picture could be different and the number of particles can be adapted to keep the $\mathrm{MSE}$ stable, see, e.g. Elvira et al. (2016).*

## 4.2 Nonasymptotic convergence

Now, in light of Proposition 2, Algorithm 1 can be understood to be solving the unconstrained optimisation problem,

$$\theta_\star := \operatorname*{arg\,min}_{\theta \in \mathbb{R}^d} V(\theta) = \operatorname*{arg\,max}_{\theta \in \mathbb{R}^d} \log p_\theta(y),\tag{11}$$

where we have access to biased and noisy gradient estimators. Below, we analyse this scheme in two distinct assumptions on the negative marginal log-likelihood $V$, first under the PŁ condition and then a strong convexity condition, utilising the results from (Demidovich et al., 2023).

For our first result, we impose the following smoothness condition, together with the PŁ condition on the marginal likelihood. For brevity we use the notation $V(\theta_*) = V_*$.

**A2.** *Assume the function $V$ is differentiable and $L$-smooth, that is, for all $(\theta, \theta') \in \mathbb{R}^d \times \mathbb{R}^d$,*

$$\|\nabla_\theta V(\theta) - \nabla_\theta V(\theta')\| \leq L \|\theta - \theta'\|,\tag{12}$$

*and also that $V$ is bounded from below by $V_\star \in \mathbb{R}$.*

**A3.** *(Polyak-Łojasiewicz) There exists a constant, $\mu > 0$, such that for all $\theta \in \mathbb{R}^d$,*

$$\|\nabla_\theta V(\theta)\|^2 \geq 2\mu(V(\theta) - V_\star).\tag{13}$$

**Theorem 1.** *Assume that $OPT(\theta_k, g_k)$, in Algorithm 1, refers to the SGD scheme outlined in (9). Under A1–A3, provided that a fixed step-size $\gamma$ is chosen such that $0 < \gamma \leq \min\{1/4L, 2/\mu\}$ we have, for every $k \in \mathbb{N}$,*

$$\mathbb{E}\left[ V(\theta_k) - V_\star \right] \leq \left( 1 - \frac{\gamma\mu}{2} \right)^k \delta_0 + \frac{8L\gamma \|\nabla_\theta U\|_\infty^2 C}{\mu N} + \frac{4 \|\nabla_\theta U\|_\infty^2 C}{\mu N},$$

*where $\delta_0 = V(\theta_0) - V_\star$ and $C = \sup_k \mathbb{E}\left[ \left( e^{A_k} \right)^2 \right] / \left( \mathbb{E}\left[ e^{A_k} \right] \right)^2 < \infty$.*

A proof can be found in Appendix B.3. Note that this provides a result on the convergence of marginal likelihood since $V(\theta) = -\log p_\theta(y)$.

**Remark 5.** *To expand this result and write it in a more intuitive form, we see that Theorem 1 essentially implies that*

$$\mathbb{E}\left[ V(\theta_k) - V_\star \right] \leq \delta_0 \left( 1 - \gamma\mu/2 \right)^k + \mathcal{O}\left( \gamma/N \right) + \mathcal{O}\left( 1/N \right).$$

*This in turn provides us a guideline on how to scale parameters, in particular, the number of iterations $k$ and the number of samples $N$. To obtain $\mathbb{E}\left[ V(\theta_k) - V_\star \right] \leq \mathcal{O}(\varepsilon)$ with $0 < \varepsilon \ll 1$, one must*

*choose $N = \mathcal{O}(\varepsilon^{-1})$ and $k \geq \mathcal{O}(\log \varepsilon^{-1})$. Notably, due to the apperance of $N$ in the second term, we do not need to take $\gamma \to 0$ for our method to converge, and rather should take $N$ large. This is in contrast to theoretical results obtained in Akyildiz et al. (2024, 2025) and Caprio et al. (2025) for alternative diffusion-based MMLE methods, where for convergence, one needs to pick $N$ large and $\gamma$ very small. See Appendix A.2 for more discussion.*

We next impose a stronger assumption on the negative log marginal likelihood to strengthen this result.

**A4.** *The following holds: $\langle \nabla V(\theta) - \nabla V(\theta'), \theta - \theta' \rangle \geq (\mu/2)\|\theta - \theta'\|^2$.*

**Remark 6.** *Note that there are practical examples where A4 is almost satisfied. For example, if $U(\theta, x)$ is strongly convex in $(\theta, x)$, then A4 can be shown to be satisfied via Prekopa-Leindler inequality (see, e.g., Saumard and Wellner (2014, Theorem 3.8)). For example, it can be shown that $U(\theta, x)$ is strictly convex in $(\theta, x)$ for the Bayesian logistic regression example, see Kuntz et al. (2023, Prop. 1).*

**Theorem 2.** *Under A1, A2, and A4, fix a step-size such that $0 < \gamma \leq \min\{1/4L, 2/\mu\}$, then we have*

$$\mathbb{E}\left[\|\theta_k - \theta^*\|^2\right] \leq \left(1 - \frac{\gamma\mu}{2}\right)^k \delta_0(2/\mu) + \frac{16L\gamma\|\nabla_\theta U\|_\infty^2 C}{\mu^2 N} + \frac{8\|\nabla_\theta U\|_\infty^2 C}{\mu^2 N},$$

*where $\delta_0 = V(\theta_0) - V_\star$ and $C = \sup_k \mathbb{E}\left[\left(e^{A_k}\right)^2\right]/\left(\mathbb{E}\left[e^{A_k}\right]\right)^2 < \infty$.*

The proof of this theorem follows from the fact that the strong convexity (A4) implies $\|\theta - \theta_\star\| \leq (2/\mu)(V(\theta) - V_\star)$, hence the bound under PL condition in Theorem 1 can be applied directly.

## 5 Experimental results

### 5.1 Bayesian logistic regression - Wisconsin cancer data

To benchmark JALA-EM's performance, relative to that of the particle gradient descent (PGD) and stochastic optimization via unadjusted Langevin algorithm (SOUL) algorithms, as introduced in Kuntz et al. (2023) and De Bortoli et al. (2021) respectively, we consider the extensively studied Bayesian logistic regression task using the Wisconsin Breast Cancer dataset, as described in De Bortoli et al. (2021). Specifically, we employ a model with a Bernoulli likelihood for the binary classification of $d_y = 683$ malignant or benign samples, and the regression weights $w \in \mathbb{R}^{d_x=9}$ are assigned an isotropic Gaussian prior $p(w|\theta, \sigma_0^2) = \mathcal{N}(w|\theta \cdot \mathbf{1}_{d_x}, \sigma_0^2)$. As is the case in Kuntz et al. (2023), we fix the prior variance to $\sigma_0^2 = 5.0$, and estimate the unique maximiser of the marginal likelihood.

To ensure a robust comparison, algorithm-specific step-sizes were tuned via a 3-fold cross-validation on the respective training-validation set, $\mathcal{D}_{train,val}$, where the Log Pointwise Predictive Density (LPPD) was utilised as the evaluation metric, since this signifies predictive performance of the fitted

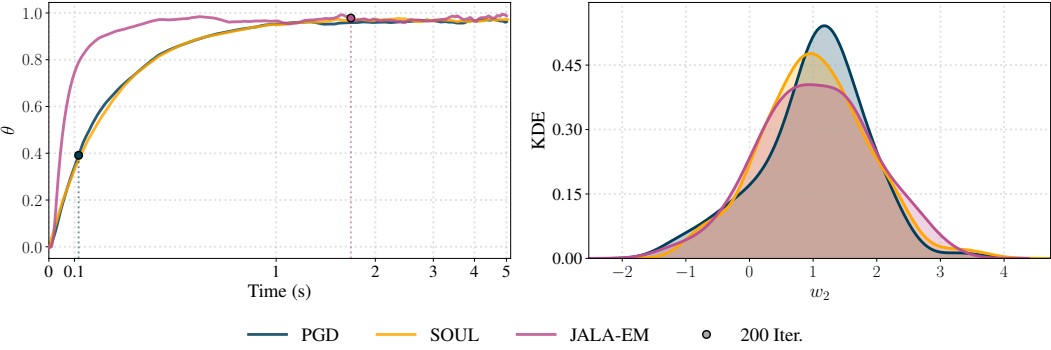

Figure 1: **Bayesian logistic regression:** Parameter estimates for PGD, SOUL, and JALA-EM, for $N = 100$ and tuned step-sizes, with a iteration milestone of 200 indicated (left). KDE of the second coordinate of the posterior approximation for the model weights (right).

model. Subsequently, final algorithm runs were constrained to a 5.0 second wall-clock duration to assess practical computational efficiency. Comprehensive experimental details can be found in Appendix C.2.

The parameter estimates and posterior approximations, illustrated in Figures 1, 5 and 6 for particle counts $N \in \{100, 50, 10\}$, highlight some of JALA-EM's distinct characteristics. Across the particle counts, JALA-EM demonstrates rapid convergence for the parameter $\theta$, consistently matching or even outperforming PGD and SOUL in terms of wall-clock time, while reaching the same estimated value. Although across algorithms all posterior approximations, for the representative regression weight $w_2$, are comparable at $N = 100$, the posterior fidelity of PGD and SOUL noticeably degrades as $N$ decreases. In contrast, JALA-EM maintains high-quality approximations, even with fewer particles, which we attribute to its effective particle management due to reweighting. Despite JALA-EM requiring an additional step-size to be tuned compared to PGD, it provides a compelling combination of fast parameter estimation and accurate posterior approximation in this setting.

## 5.2 Error model selection - Bayesian regression

To demonstrate JALA-EM's unique and dual capability to concurrently estimate both parameters and the marginal likelihood, we consider a Bayesian model selection problem. Specifically, we aim to differentiate between two competing, nested Bayesian linear regression models, in which one is the true generating process $\mathcal{G}$ of the synthetic dataset $\mathcal{D}$, with model preference quantified via Bayes Factors derived from these marginal likelihood estimates. Specifically, these models aim to capture the relationship $y_i = X_i w + \varepsilon_i$ through a latent weight vector $w \in \mathbb{R}^{d_x}$, where $\varepsilon_i$ represents the observation error. The first model, $\mathcal{M}_G$, assumes i.i.d. Gaussian observation errors, $\varepsilon_i \sim \mathcal{N}(0, \sigma^2)$, while the second, $\mathcal{M}_T$, assumes Student-t distributed errors, $\varepsilon_i \sim \text{Student-t}(0, \sigma^2, \nu)$, providing robustness against outliers. For both models, an isotropic Gaussian prior is placed on the regression weights $w$, $p(w|\alpha) = \mathcal{N}(0, \alpha^{-1} I_{d_x})$. Notably, for numerical stability and to ensure positivity, we use JALA-EM to estimate the log-transformed model parameters, $\phi_1 = \log \sigma^2$, $\phi_1 = \log \alpha$, and for $\mathcal{M}_T$, $\phi_3 = \log \nu$.

Indeed, to estimate the log marginal likelihood $\log \hat{Z}_{\mathcal{M}}$, for model $\mathcal{M}$, we have, by Proposition 1, that after $K$ steps

$$\log \hat{Z}_{\mathcal{M},K} = \log Z_{\mathcal{M},0} + \log \left( \sum_{i=1}^{N} e^{A_K^i} \right) - \log N \approx \log p(y|X, \theta_{\mathcal{M},K}, \mathcal{M}), \qquad (14)$$

where $Z_{\mathcal{M},0}$ is the marginal likelihood evaluated at the initial estimates, $\theta_{\mathcal{M},0}$. Furthermore, we adopt the following version of (14) which includes resampling as usual in SMC contexts, cfr. Del Moral et al. (2006):

$$\log \hat{Z}_{\mathcal{M},K} = \log Z_{\mathcal{M},0} + \sum_{j=1}^{J} \log \left( \frac{1}{N} \sum_{i=1}^{N} \exp \left( \sum_{m=k_{j-1}+1}^{k_j} A_m^i \right) \right), \quad k_0 = 1 \qquad (15)$$

where $k_j$ is the step at which the $j$-th resampling event occurs, with $j = 1, \ldots, J$, whilst $A_m^i$ is the log-incremental weight of particle $i$ at step $m$, and $\sum_{m=k_{j-1}+1}^{k_j} A_m^i$ represents the cumulative log-weight of particle $i$ between two resampling steps.

Notably, the computation of the marginal likelihood is model dependent, as for $\mathcal{M}_G$ this is analytically tractable, whereas for $\mathcal{M}_T$ it is generally intractable, and thus we estimate it using Importance Sampling, with a proposal distribution derived from a Gaussian approximation to the posterior of $w$ using initial parameter guesses. Comprehensive experimental details and corresponding derivations can be found in Appendix C.4.

Our numerical experiments focus on generating $\mathcal{D}$ with $d_y = 500$ and $d_x = 8$, with features drawn from $\mathcal{N}(0, I_{d_x})$. True parameters are set to $\alpha^* = 1.0$, $\sigma^* = 1.0$, and $\nu^* = 4$ (for $\mathcal{G} = \mathcal{M}_T$), with $w_*$ drawn from $\mathcal{N}(0, \alpha_*^{-1} I_{d_x})$. This core experimental trial was repeated 100 times, with JALA-EM configured with $N = 50$ and $K = 250$. The results, exemplified by Figure 7, highlight rapid convergence to sensible parameter estimates and estimated marginal likelihoods clearly differentiating between the true and misspecified models. Notably, when $\mathcal{G} = \mathcal{M}_G$, $\log \hat{Z}_{\mathcal{M}_G,K}$ quickly stabilises close to its true analytical value, and precisely matches the analytical marginal likelihood derived

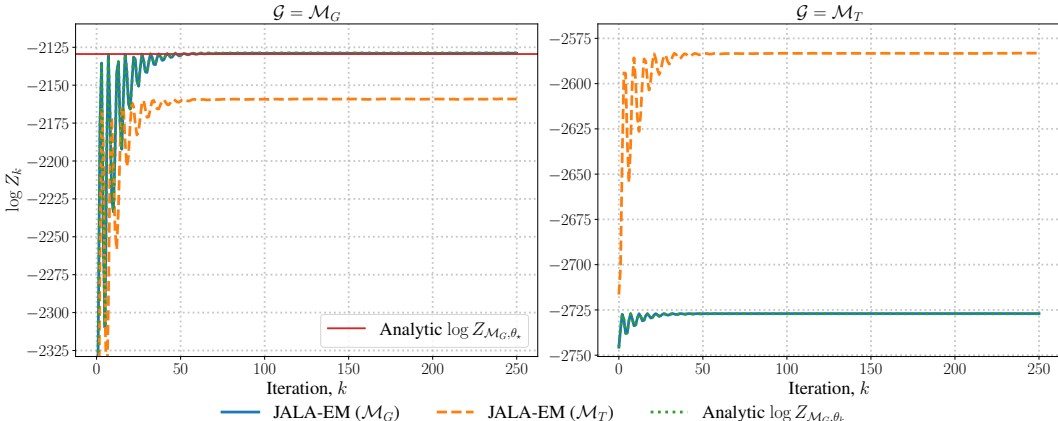

Figure 2: **Error model selection:** Marginal likelihood estimates for JALA-EM fitting $\mathcal{M}_G$ and $\mathcal{M}_T$, where the true underlying model is the former (left) and the latter (right). Here, $d_y = 1500$, $d_x = 8$, and $N = 50$. Parameter estimates are initialised at values perturbed away from the true values.

from the iterated parameter estimates, corroborating Proposition 1. Additionally, when $\mathcal{G} = \mathcal{M}_T$, JALA-EM fitting $\mathcal{M}_T$ is more effective at recovering the true parameter values, and importantly its $\log \hat{Z}_{\mathcal{M}_T, \theta_k}$ stabilises at a higher value. In fact, in Figure 2, where $d_y = 1500$, JALA-EM's ability to effectively leverage larger datasets for more confident model selection is highlighted, consistent with Bayesian principles. Lastly, we note that, across the 100 repetitions, model selection based on the higher estimated marginal likelihood correctly identified the true underlying model class in 100% of trials when $\mathcal{G} = \mathcal{M}_G$, and 99% of trials when $\mathcal{G} = \mathcal{M}_T$, indicating JALA-EM's robustness and utility for model selection.

## 5.3 Model order selection - Bayesian regression

A related, but distinct problem is that of model order selection in the context of competing, nested Bayesian polynomial regression models. In this case, the candidate models share the same probabilistic structure for both their weights and observation errors, and instead differ in the model specific features utilised, which are derived from the polynomial basis expansion up to and including order $p$, denoted by $\varphi_p(x_i)$, for the $p$-th model $\mathcal{M}_p$. To be clear, these models aim to capture the relationship $y_i = \varphi_p(x_i)^\top w_p + \varepsilon_i$, through a latent vector $w_p \in \mathbb{R}^{p+1}$, where $\varepsilon_i \sim \mathcal{N}(0, \sigma^2)$.

Following the comprehensive setup described in Appendix C.5, we compare the performance of JALA-EM to that of a competitive baseline, which notably leverages Ordinary Least Squares (OLS) for weight estimation followed by Bayesian Information Criterion (BIC) for model selection, over a range of true model orders $p_\star \in [2, 8]$, where we repeat the core experimental trial 100 times for each order. As seen in Table 1, we observe JALA-EM to match or outperform the aforementioned baseline across the values of $p_\star$ and $d_y$ considered, where outperformance is notably more pronounced for higher model orders. We attribute this to JALA-EM leveraging an approximation of an integral over the latent parameters, which yields a more robust trade-off between model complexity and fit than the baseline's reliance on a point estimate and BIC approximation, particularly in the non-asymptotic settings considered.

## 5.4 Bayesian neural network

As motivated in Kuntz et al. (2023), Bayesian neural network (BNN) models represent a more complex task due to the typically multimodal nature of their posterior distributions. In this setting, we first consider a BNN for the binary classification task of distinguishing between the MNIST handwritten digits 4 and 9, before extending this to a multi-class setting distinguishing between the digits 2, 4, 7, and 9, for a variety of BNN model capacities. Full experimental details are provided in Appendix C.3.

Following the setup of Kuntz et al. (2023), we observe short transient phases in the evolution of parameter estimates—albeit converging to different local maxima—as well as in evaluation metrics

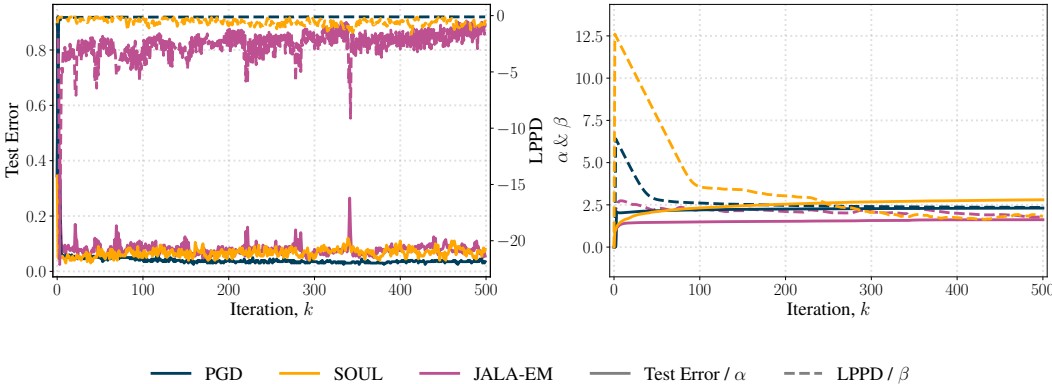

Figure 3: **Bayesian neural network:** LPPD and Test Error, for PGD, SOUL, and JALA-EM particle clouds, with $N = 100$ and a global step-size of 0.1, as in Kuntz et al. (2023) (left). Parameter estimates, $(\alpha, \beta) = (\log(\sigma_w), \log(\sigma_v))$, where $\sigma_w$ and $\sigma_v$ are the variances of the zero-mean isotropic Gaussian priors applied to the weights $W$ and $V$ (right).

for PGD, SOUL, and JALA-EM, as illustrated in Figure 3. Notably, the predictive performance of JALA-EM is comparable to that of SOUL across all experiments, which is to be expected due to the correlations that exist between particles due to the resampling mechanism present in JALA-EM. Although PGD converges faster in terms of wall-clock time, it does not support marginal likelihood estimation, which is a key advantage of our method. Overall, this example highlights the reasonable scalability of our proposal to more complex and high-dimensional inference problems.

## 6    Conclusions

We provided an SMC-based EM algorithm that exploits unadjusted Langevin dynamics together with SMC corrections to implement a MMLE procedure. Our method is general and can be interfaced with different optimisation methods for efficiency. One distinct feature of our method is the ability to compute normalising constants on-the-fly, hence the ability to select models during training runs. We demonstrate the performance of our method on a variety of regression examples, as well as on a Bayesian neural network example.

**Limitations and future work.** Besides SOUL and PGD, other benchmarks (such as using Metropolis-Hastings for the E-step or gradient-free optimisation for the M-step) would be interesting to consider. Additionally, our work can use a large number of computational tricks from particle filtering literature, since our method is closely related. Our method is SMC-based, hence there are computational limitations due to weight computations. Without a careful implementation, one can run into weight degeneracy issues as is common in SMC, especially in high-dimensions, see, e.g., Bickel et al. (2008). One can resort to alternative strategies, such as tempering (Crucinio, 2025) or weight clipping (Martino et al., 2018) to improve the performance.

**Broader impact.** Our method provides a general purpose training scheme for fitting LVMs which are ubiquitous in science and engineering. Our methodology can help faster and more reliable model testing in a variety of scenarios, accelerating the process of model development and selection for a range number of applications. However, as a standard training algorithm, it could be also used to train models for harmful activities, but this risk is not any larger than a standard training method.

## Acknowledgements

J. C. is supported by EPSRC through the Modern Statistics and Statistical Machine Learning (StatML) CDT programme, grant no. EP/S023151/1. D.C. worked under the auspices of Italian National Group of Mathematical Physics (GNFM) of INdAM. D.C. expresses their gratitude to Marylou Gabrié for the support. O. D. A. is grateful to Department of Statistics, London School of Economics and Political Science (LSE) for the hospitality during the preparation of this work. We thank anonymous referees for their insightful feedback which improved our work.

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

## A Discussion and comparisons

In this section, we put our method in a more general context by comparing it to the recently proposed body of interacting particle methods for the MMLE problem cited throughout the paper. We will compare in this section, in particular, particle gradient descent (PGD) of Kuntz et al. (2023) and interacting particle Langevin algorithm (IPLA) of Akyildiz et al. (2025), as well as the slow-fast Langevin algorithm (SFLA) method discussed in Akyildiz et al. (2024).

### A.1 Algorithmic comparison

Our method is closely related to PGD and interacting particle Langevin algorithm (IPLA) which rely on particle systems to update the parameter iterates. To be specific, given some initialisation $(\theta_0, X_0^1, \ldots, X_0^N)$, PGD (Kuntz et al., 2023) consists of the following discrete dynamical system:

$$\theta_{k+1} = \theta_k - \frac{\gamma}{N} \sum_{i=1}^{N} \nabla_\theta U(\theta_k, X_k^i), \tag{16}$$

$$X_{k+1}^i = X_k^i - \gamma \nabla_x U(\theta_k, X_k^i) + \sqrt{2\gamma} Z_{k+1}^i, \tag{17}$$

where $Z_{k+1}^i \sim \mathcal{N}(0, I_d)$ are independent standard Gaussian random variables. The IPLA method of Akyildiz et al. (2025) is very similar, with a scaled noise in $\theta$ iterates:

$$\theta_{k+1} = \theta_k - \frac{\gamma}{N} \sum_{i=1}^{N} \nabla_\theta U(\theta_k, X_k^i) + \sqrt{\frac{2\gamma}{N}} \xi_{k+1}, \tag{18}$$

$$X_{k+1}^i = X_k^i - \gamma \nabla_x U(\theta_k, X_k^i) + \sqrt{2\gamma} Z_{k+1}^i, \tag{19}$$

where $\xi_{k+1} \sim \mathcal{N}(0, I_{d_\theta})$ is another independent standard Gaussian random variable. In the above equations, one can see parameters updates (16) and (18) play a similar role to our optimiser steps in (9). However, in this setting, particles are *equally weighted* $(1/N)$, and the latent variables are updated through a ULA step in (17) and (19) similar to our case. One crucial aspect for both methods is that the step-size $\gamma$ needs to be same for both $\theta$ and $X$ updates as these algorithms are derived from an Euler discretisation of a continuous-time dynamics. This is a significant limitation in practice as the scales of $\theta$ and $X$ can be very different, and thus, for example, the step-size $\gamma$ needs to be chosen very small to ensure stability of the $X$ updates, which in turn leads to very slow updates for $\theta$. A related aspect here is that while, for example, Kuntz et al. (2023) uses an adaptive optimiser for $\theta$-updates in challenging experiments, theoretical results for PGD (Caprio et al., 2025) and IPLA are both tied to the use of plain gradient updates as in (16) and (18). In contrast, our method allows for different step-sizes for $\theta$ and $X$ updates, as well as the use of adaptive optimisers. This is because our method is not derived from a discretisation of a continuous-time dynamics, but rather from a direct optimisation of the MMLE objective.

To address some of the limitations above, the continuous-time dynamics behind PGD and IPLA have been extended in Akyildiz et al. (2024); Oliva et al. (2025) to allow for different time-scales for $\theta$ and $X$ updates. For example, the resulting slow-fast Langevin algorithm (SFLA) method of Akyildiz et al. (2024) consists of the following updates:

$$\theta_{k+1} = \theta_k - \gamma \nabla_\theta U(\theta_k, X_k) + \sqrt{\frac{2\gamma}{\beta}} \xi_{k+1}, \tag{20}$$

$$X_{k+1} = X_k - \frac{\gamma}{\epsilon} \nabla_x U(\theta_k, X_k) + \sqrt{\frac{2\gamma}{\epsilon}} Z_{k+1}, \tag{21}$$

where $\beta > 0$ is an inverse temperature parameter, $\epsilon > 0$ is a scale parameter, and $\gamma > 0$ is a step-size. Here, there is a single latent variable $X_k$ which is updated with a step-size $\gamma/\epsilon$ that can be much larger than the step-size $\gamma$ used for $\theta$ updates (see Oliva et al. (2025) for a related particle-based approach). However, this comes at the cost of introducing two additional hyperparameters $\beta$ and $\epsilon$, which can be difficult to tune in practice. Moreover, as for PGD and IPLA, theoretical results for SFLA are tied to the use of plain gradient updates as in (20), and do not allow for adaptive optimisers.

## A.2 Theoretical comparison

Both PGD and IPLA come with nonasymptotic bounds; see the analysis in Caprio et al. (2025) and Akyildiz et al. (2025). Under strong convexity and gradient smoothness assumptions on $U$, both methods provide theoretical guarantees of the following form:

$$\mathbb{E}[\|\theta_k - \theta_\star\|^2]^{1/2} \leq \mathcal{O}(\gamma^{1/2} + N^{-1/2} + e^{-\gamma\mu k}), \tag{22}$$

where $\mu > 0$ is a strong convexity constant, $\gamma$ is the step-size, $N$ is the number of particles, and $\theta_\star$ is the unique minimiser of the MMLE objective. One can see that, in addition to the fact that these methods have to work with the same step-size $\gamma$ for both $\theta$ and $X$ updates, the above bound also requires $\gamma$ to be very small to control the bias term $\mathcal{O}(\gamma^{1/2})$. Specifically, for the error to vanish, one needs take large $N$ and small $\gamma$. On the other hand, the convergence rate is governed by the exponential term $\mathcal{O}(e^{-\gamma\mu k})$, which decays faster for larger $\gamma$, resulting in a trade-off between accuracy and convergence speed.

In contrast, our result in Theorem 2 provides an error bound:

$$\mathbb{E}[\|\theta_k - \theta_\star\|^2]^{1/2} \leq \mathcal{O}\left((1 - \gamma\mu/2)^{k/2} + \gamma^{1/2}N^{-1/2} + N^{-1/2}\right). \tag{23}$$

In the above display, it suffices to take $N$ large for error to be small. The step-size $\gamma$ can be chosen independently of $N$ and only affects the convergence speed, thus decoupling the accuracy and convergence speed trade-off.

Our results can also be extended to adaptive optimisers, see, e.g. Surendran et al. (2024) whereas theoretical results for PGD, IPLA, and SFLA are tied to the use of plain gradient updates.

## A.3 General comparison of the methods

We provide the following table summarising the main differences between the methods discussed above.

| Feature | PGD | SOUL | SFLA | JALA-EM |
|---|---|---|---|---|
| Marginal likelihood estimation | ✗ | ✗ | ✗ | ✓ |
| Different $\theta$ and $x$ step-sizes | ✗ | ✓ | ✓ | ✓ |
| Theoretical guarantees with fixed $\gamma$ | ✓ | ✗ | ✓ | ✓ |
| Adaptation of theory for adaptive optimizers | ✗ | ✗ | ✗ | ✓ |

# B Theoretical results

## B.1 Proof of Proposition 1

*Proof.* This proof follows mutatis mutandis from the proof of Proposition 1 in Carbone et al. (2023), we include here for completeness.

First, note that under the ULA update given in (2), the transition probability density, that is, the probability density of moving from $x$ to $y$ at time $k$, is given by

$$\beta_k(x, y) = (4\pi h)^{-d/2} \exp\left(-\frac{1}{4h}|y - x + h\nabla U_k(x)|^2\right),$$

We also observe that:

$$A_k = \sum_{q=1}^{k} [\alpha_{q-1}(X_{q-1}, X_q) - \alpha_q(X_q, X_{q-1})], \quad k \in \mathbb{N}. \tag{24}$$

Now, the need for this transition probability density becomes apparent when attempting to calculate expectations over the law of the joint process, as we specifically have, for a scalar test function $\varphi$

$$\mathbb{E}\left[\varphi(X_k)e^{A_k}\right] = \int_{\mathbb{R}^{d \cdot k}} \varphi(x_k)e^{A_k}\rho(x_0, \ldots, x_k)dx_0 \cdots dx_k, \tag{25}$$

where $\rho(x_0, \ldots, x_k)$ denotes the joint probability density function of the path $(X_0, \ldots, X_k)$ at $k \in \mathbb{N}$, and is defined as

$$\rho(x_0, \ldots, x_k) = \pi_0(x_0) \prod_{q=0}^{k-1} \beta_q(x_q, x_{q+1}),$$

$$= \pi_0(x_0) \prod_{q=1}^{k} \beta_{q-1}(x_{q-1}, x_q),$$

In light of this, it remains to express $e^{A_k}$ in terms of the transition probability density, $\beta$. Indeed, we have from (2) that

$$A_k = \sum_{q=1}^{k} \left[ \alpha_{q-1}(X_{q-1}, X_q) - \alpha_q(X_q, X_{q-1}) \right], \quad k \in \mathbb{N}. \tag{26}$$

Thus, by substituting (3) into (26), we have that

$$\exp(A_k) = \prod_{q=1}^{k} \exp\left( \alpha_{q-1}(X_{q-1}, X_q) - \alpha_q(X_q, X_{q-1}) \right),$$

$$= \prod_{q=1}^{k} \exp\left( U_{q-1}(X_{q-1}) + \frac{1}{2}(X_q - X_{q-1}) \cdot \nabla U_{q-1}(X_{q-1}) + \frac{h}{4}|\nabla U_{q-1}(X_{q-1})|^2 \right.$$

$$\left. - U_q(X_q) - \frac{1}{2}(X_{q-1} - X_q) \cdot \nabla U_q(X_q) - \frac{h}{4}|\nabla U_q(X_q)|^2 \right),$$

$$= \exp(U_0(X_0) - U_k(X_k)) \prod_{q=1}^{k} \exp\left( \frac{1}{2}(X_q - X_{q-1}) \cdot \nabla U_{q-1}(X_{q-1}) + \frac{h}{4}|\nabla U_{q-1}(X_{q-1})|^2 \right.$$

$$\left. - \frac{1}{2}(X_{q-1} - X_q) \cdot \nabla U_q(X_q) - \frac{h}{4}|\nabla U_q(X_q)|^2 \right),$$

$$= \exp(U_0(X_0) - U_k(X_k)) \prod_{q=1}^{k} \frac{\beta_q(X_q, X_{q-1})}{\beta_{q-1}(X_{q-1}, X_q)},$$

where in the last equality we have used the fact that

$$\frac{\beta_q(X_q, X_{q-1})}{\beta_{q-1}(X_{q-1}, X_q)} = \frac{(4\pi h)^{-\frac{d}{2}} \exp\left[ -\frac{1}{4h}|X_{q-1} - X_q + h\nabla U_q(X_q)|^2 \right]}{(4\pi h)^{-\frac{d}{2}} \exp\left[ -\frac{1}{4h}|X_q - X_{q-1} + h\nabla U_{q-1}(X_{q-1})|^2 \right]} \tag{27}$$

$$= \exp\left( \frac{1}{2}(X_q - X_{q-1}) \cdot \nabla U_{q-1}(X_{q-1}) + \frac{h}{4}|\nabla U_{q-1}(X_{q-1})|^2 \right.$$

$$\left. - \frac{1}{2}(X_{q-1} - X_q) \cdot \nabla U_q(X_q) - \frac{h}{4}|\nabla U_q(X_q)|^2 \right). \tag{28}$$

as we expanded the squares at numerator and denominator and we have notably leveraged the fact that $\frac{1}{4h}|X_q - X_{q-1}|^2 - \frac{1}{4h}|X_{q-1} - X_q|^2 = 0$.

Now, recalling that $Z_k = \int e^{-U_k(x)} dx$ and bringing everything together into 25, we obtain

$$
\begin{aligned}
\mathbb{E}\left[\varphi(X_k)e^{A_k}\right] &= \int_{\mathbb{R}^{d \cdot k}} \varphi(x_k)e^{A_k}\rho(x_0, \dots, x_k)\, dx_0 \cdots dx_k, \\
&= \int_{\mathbb{R}^{d \cdot k}} \varphi(x_k)\exp\left(U_0(x_0) - U_k(x_k)\right)\prod_{q=1}^{k}\beta_q(x_q, x_{q-1}) \cdot p_0(x_0)\ dx_0 \cdots dx_k, \\
&= \frac{1}{Z_0}\int_{\mathbb{R}^{d \cdot k}} \varphi(x_k)\exp\left(-U_k(x_k)\right)\prod_{q=1}^{k}\beta_q(x_q, x_{q-1})\, dx_0 \cdots dx_k, \\
&= \frac{1}{Z_0}\int_{\mathbb{R}^{d}} \varphi(x_k)\exp\left(-U_k(x_k)\right)\ dx_k.
\end{aligned}
$$

where in the last equality we have have iteratively integrated over $x_0$, $x_1$, up to and including $x_{k-1}$, where we note that, for all $q \in \mathbb{N}$ and all $x_{q-1} \in \mathbb{R}^d$,

$$
\int_{\mathbb{R}^d} \beta_q(x_q, x_{q-1})dx_{q-1} = 1.
$$

Thus, if we set $\varphi(X_k) = 1$, we obtain

$$
\mathbb{E}\left[e^{A_k}\right] = \frac{1}{Z_0}\int_{\mathbb{R}^d}\exp\left(-U_k(x_k)\right)dx_k = \frac{Z_k}{Z_0},
$$

This completes the proof, since $\mathbb{E}\left[\varphi(X_k)e^{A_k}\right]/\mathbb{E}\left[e^{A_k}\right] = \mathbb{E}_{\pi_k}\left[\varphi(X_k)\right]$, and as generalization of this result to vector-valued $\varphi$ is straightforward. $\square$

See Carbone (2025) for a further insight on different choices of the transition kernel $\beta(x, y)$.

## B.2 Proof of Proposition 2

Now, we derive an MSE bound on the error introduced through utilising the sample based approximation

$$
\frac{\mathbb{E}\left[\nabla_\theta U(\theta_k, X_k)e^{A_k}\right]}{\mathbb{E}\left[e^{A_k}\right]} \approx \frac{\sum_{i=1}^{N}\nabla_\theta U(\theta_k, X_k^{(i)})e^{A_k^{(i)}}}{\sum_{i=1}^{N}e^{A_k^{(i)}}},
$$

where, to be clear, the expectations on the left-hand side are over the law of the joint process $(X_k, A_k)$, as defined through (2). Throughout, we assume that $\{(X_k^{(i)}, A_k^{(i)})\}_{i=1}^{N}$ are i.i.d copies of $(X_k, A_k)$, or equivalently that no resampling is performed up to time $k$.

Specifically, we wish to bound the quantity

$$
\mathbb{E}\left[\left\|\frac{\mathbb{E}\left[\nabla_\theta U(\theta_k, X_k)e^{A_k}\right]}{\mathbb{E}\left[e^{A_k}\right]} - \frac{\sum_{i=1}^{N}\nabla_\theta U\left(\theta_k, X_k^{(i)}\right)e^{A_k^{(i)}}}{\sum_{i=1}^{N}e^{A_k^{(i)}}}\right\|^2\right] = \mathbb{E}\left[\left\|\frac{\mathbb{E}[W_k F_k]}{\mathbb{E}[W_k]} - \frac{\overline{WF}_k}{\overline{W}_k}\right\|^2\right],
$$

where we have let $W_k = e^{A_k}$ and $F_k = \nabla_\theta U(\theta_k, X_k)$ for notational brevity, whilst sample based approximations, $\overline{W}_k = \frac{1}{N}\sum_{i=1}^{N}e^{A_k^{(i)}}$ and $\overline{WF}_k = \frac{1}{N}\sum_{i=1}^{N}\nabla_\theta U(\theta_k, X_k^{(i)})e^{A_k^{(i)}}$, are indicated by an overset bar.

We follow the proof of Theorem 1 in Akyildiz and Míguez (2021). First, we note that the following inequality holds,

$$
\left\| \frac{\mathbb{E}\left[\nabla_\theta U(\theta_k, X_k)e^{A_k}\right]}{\mathbb{E}\left[e^{A_k}\right]} - \frac{\sum_{i=1}^N \nabla_\theta U(\theta_k, X_k^{(i)})e^{A_k^{(i)}}}{\sum_{i=1}^N e^{A_k^{(i)}}} \right\| = \left\| \frac{\mathbb{E}[W_k F_k]}{\mathbb{E}[W_k]} - \frac{\overline{WF}_k}{\overline{W}_k} \right\|,
$$

$$
= \frac{\left\| \mathbb{E}[W_k F_k]\,\overline{W}_k - \overline{WF}_k\,\mathbb{E}[W_k] \right\|}{|\mathbb{E}[W_k]|\,|\overline{W}_k|},
$$

$$
= \frac{\left\| \overline{W}_k(\mathbb{E}[W_k F_k] - \overline{WF}_k) + \overline{WF}_k(\overline{W}_k - \mathbb{E}[W_k]) \right\|}{|\mathbb{E}[W_k]|\,|\overline{W}_k|},
$$

$$
\leq \frac{|\overline{W}_k|\,\left\| \mathbb{E}[W_k F_k] - \overline{WF}_k \right\| + \left\| \overline{WF}_k \right\|\,|\overline{W}_k - \mathbb{E}[W_k]|}{|\mathbb{E}[W_k]|\,|\overline{W}_k|},
$$

$$
= \frac{\left\| \mathbb{E}[W_k F_k] - \overline{WF}_k \right\|}{|\mathbb{E}[W_k]|} + \left\| \overline{WF}_k \right\|\,\left| \frac{1}{\mathbb{E}[W_k]} - \frac{1}{\overline{W}_k} \right|,
$$

$$
\leq \frac{\left\| \mathbb{E}[W_k F_k] - \overline{WF}_k \right\|}{|\mathbb{E}[W_k]|} + \|F_k\|_\infty\,|\overline{W}_k|\,\left| \frac{\overline{W}_k - \mathbb{E}[W_k]}{\mathbb{E}[W_k]\overline{W}_k} \right|,
$$

$$
= \frac{\left\| \mathbb{E}[W_k F_k] - \overline{WF}_k \right\|}{\mathbb{E}[W_k]} + \|F_k\|_\infty \frac{|\overline{W}_k - \mathbb{E}[W_k]|}{\mathbb{E}[W_k]},
$$

where in the first inequality we have repeatedly applied the triangle inequality, and in the fourth equality simply divided the denominator through each term. Furthermore, in the second inequality we have made use of the fact that $\sum_{i=1}^N \nabla_\theta U(\theta_k, X_k^{(i)})e^{A_k^{(i)}} \leq \|\nabla_\theta U\|_\infty \sum_{i=1}^N e^{A_k^{(i)}}$, recalling that we assume $\|\nabla_\theta U\|_\infty < \infty$, for any $\theta \in \mathbb{R}^p$ and $X \in \mathbb{R}^d$.

Now, utilising the fact that $(a+b)^2 \leq 2(a^2 + b^2)$, and taking the expectation of both sides, we obtain

$$
\mathbb{E}\left[ \left\| \frac{\mathbb{E}[W_k F_k]}{\mathbb{E}[W_k]} - \frac{\overline{WF}_k}{\overline{W}_k} \right\|^2 \right] \leq \mathbb{E}\left[ \left( \frac{\left\| \mathbb{E}[W_k F_k] - \overline{WF}_k \right\|}{\mathbb{E}[W_k]} + \|F_k\|_\infty \frac{|\overline{W}_k - \mathbb{E}[W_k]|}{\mathbb{E}[W_k]} \right)^2 \right],
$$

$$
\leq \frac{2\mathbb{E}\left[ \| \mathbb{E}[W_k F_k] - \overline{WF}_k \|^2 \right]}{(\mathbb{E}[W_k])^2} + 2\|F_k\|_\infty^2 \frac{\mathbb{E}\left[ (\overline{W}_k - \mathbb{E}[W_k])^2 \right]}{(\mathbb{E}[W_k])^2},
$$

$$
= \frac{2}{N} \frac{\left( \mathbb{E}[\|W_k F_k\|^2] - \|\mathbb{E}[W_k F_k]\|^2 \right)}{(\mathbb{E}[W_k])^2} + \frac{2\|F_k\|_\infty^2}{N} \frac{\left( \mathbb{E}[W_k^2] - (\mathbb{E}[W_k])^2 \right)}{(\mathbb{E}[W_k])^2},
$$

$$
\leq \frac{2}{N} \frac{\mathbb{E}[\|W_k F_k\|^2]}{(\mathbb{E}[W_k])^2} + \frac{2\|F_k\|_\infty^2}{N} \frac{\mathbb{E}[W_k^2]}{(\mathbb{E}[W_k])^2},
$$

$$
\leq \frac{2\|F_k\|_\infty^2}{N} \frac{\mathbb{E}[W_k^2]}{(\mathbb{E}[W_k])^2} + \frac{2\|F_k\|_\infty^2}{N} \frac{\mathbb{E}[W_k^2]}{(\mathbb{E}[W_k])^2},
$$

$$
= \frac{4\|F_k\|_\infty^2}{N} \frac{\mathbb{E}[W_k^2]}{(\mathbb{E}[W_k])^2}.
$$

giving the desired result, where in the first equality we have leveraged the fact that the numerators, on the right-hand side, are perfect Monte Carlo estimates, and so this equality holds through a standard variance decomposition. We also note that in the second inequality we have again used the fact that $F = \nabla_\theta U(\theta_k, X_k)$ is bounded by $\|F\|_\infty = \|\nabla_\theta U\|_\infty$, so that

$$
\|W_k F_k\|^2 \leq \|F_k\|_\infty^2 W_k^2 \quad \Rightarrow \quad \mathbb{E}[\|W_k F_k\|^2] \leq \|F_k\|_\infty^2 \mathbb{E}[W_k^2],
$$

$$
\|W_k F_k\| \leq \|F_k\|_\infty |W_k| \quad \Rightarrow \quad \|\mathbb{E}[W_k F_k]\| \leq \mathbb{E}[\|W_k F_k\|] \leq \|F_k\|_\infty \mathbb{E}[|W_k|].
$$

To conclude, we exploit assumption **A**1

$$
\frac{4\|F_k\|_\infty^2}{N} \frac{\mathbb{E}[W_k^2]}{(\mathbb{E}[W_k])^2} \leq \frac{4\|F_k\|_\infty^2}{N} \sup_k \frac{\mathbb{E}[W_k^2]}{(\mathbb{E}[W_k])^2}. \tag{29}
$$

## B.3 Proof of Theorem 1

We specifically leverage Theorem 4 from Demidovich et al. (2023), ensuring that the respective assumptions are satisfied.

**A5.** *There exist constants, $A, B, C, b, c \geq 0$, such that the gradient estimator, $g(\theta)$, for every $\theta \in \mathbb{R}^d$ satisfies*

$$\langle \nabla V(\theta), \mathbb{E}[g(\theta)] \rangle \geq b \|\nabla V(\theta)\|^2 - c, \tag{30}$$

$$\mathbb{E}\left[\|g(\theta)\|^2\right] \leq 2A(V(\theta) - V^*) + B\|\nabla V(\theta)\|^2 - C. \tag{31}$$

*where $V_* \equiv V(\theta_*)$.*

**Theorem 3.** *Let Assumptions A2, A3, and A5 hold. Choose a step size, $\gamma$, such that*

$$0 < \gamma < \min\left\{\frac{\mu b}{L(A + \mu B)}, \frac{1}{\mu b}\right\}, \tag{32}$$

*then we have, for every $k \in \mathbb{N}$,*

$$\mathbb{E}\left[V(\theta_k) - V^*\right] \leq (1 - \gamma \mu b)^k \delta_0 + \frac{LC\gamma}{2\mu b} + \frac{c}{\mu b}, \tag{33}$$

*where $\delta_0 = V(\theta_0) - V^*$.*

In light of Assumption A5 lacking an intuitive explanation, we turn to Assumption 7 of Demidovich et al. (2023):

**A6.** *For all $\theta \in \mathbb{R}^d$, there exists $\Delta \geq 0$ such that*

$$\mathbb{E}\left[\|g(\theta) - \nabla V(\theta)\|^2\right] \leq \Delta^2, \tag{34}$$

We remark that A6 is notably a stronger assumption than that of A5, and through Theorem 13 of Demidovich et al. (2023) we have that Assumption A6 implies Assumption A5 with $A = 0$, $B = 2$, $C = 2\Delta^2$, $b = \frac{1}{2}$, and $c = \frac{\Delta^2}{2}$.

Critically, if the assumptions of Proposition 2 hold, we have that

$$\mathbb{E}\left[\|\nabla_\theta V(\theta_k) - g(\theta_k)\|^2\right] \leq \Delta^2; \quad \Delta^2 = \frac{4C\|\nabla_\theta U\|_\infty^2}{N}, \tag{35}$$

where $C = \sup_k \mathbb{E}\left[\left(e^{A_k}\right)^2\right] / \left(\mathbb{E}\left[e^{A_k}\right]\right)^2$ and so Assumption 7 of Demidovich et al. (2023) is satisfied for this $\Delta$.

Thus, if Assumptions A2 and A3 hold, as well as those of Proposition 2, we have that, for a step size

$$0 < \gamma < \min\left\{\frac{\mu b}{L(A + \mu B)}, \frac{1}{\mu b}\right\} = \min\left\{\frac{\frac{\mu}{2}}{L(0 + 2\mu)}, \frac{1}{\frac{\mu}{2}}\right\},$$

$$= \min\left\{\frac{1}{4L}, \frac{2}{\mu}\right\},$$

for every $k \in \mathbb{N}$,

$$\mathbb{E}\left[V(\theta_k) - V^*\right] \leq (1 - \gamma \mu b)^k \delta_0 + \frac{LC\gamma}{2\mu b} + \frac{c}{\mu b},$$

$$= (1 - \gamma \frac{\mu}{2})^k \delta_0 + \frac{2\Delta^2 L\gamma}{2\frac{\mu}{2}} + \frac{\frac{\Delta^2}{2}}{\frac{\mu}{2}},$$

$$= (1 - \frac{\gamma\mu}{2})^k \delta_0 + \frac{8L\gamma\|\nabla_\theta U\|_\infty^2}{\mu N}C + \frac{4\|\nabla_\theta U\|_\infty^2}{\mu N}C,$$

where $\delta_0 = V(\theta_0) - V^*$.

To conclude, we note that $V(\theta_k) - V^* = \log p_{\theta*}(y) - \log p_{\theta_k}(y)$.

## C Experimental details

### C.1 Implementation details

In this section, we outline the details of implementation for our method.

The resampling step in Algorithm 1 proceeds as follows. Given that it is triggered at step $k_r$, first we resample the walkers $X_i^{k_r}$ using the normalized weights $w_i^{k_r}$ as probability of picking the $i$-th particle and we reset $A_i^{k_r} = 0$. Several routines for resampling have been developed in the context of particle filtering, cfr. Douc and Cappé (2005); Li et al. (2015). In our experiments we decided to adopt systematic resampling, since not only is it computationally efficient, with order $\mathcal{O}(N)$, but also the variance of the number of resampled particles is reduced, compared to that of say stratified resampling (Li et al., 2015). As described in Carbone et al. (2023), an extra stage is then necessary to track $Z_k$ through the resampling step, that is adopting

$$Z_k = Z_{k_r} \frac{1}{N} \sum_{i=1}^{N} e^{A_k^i}$$

for all $k \geq k_r$ until the next resampling step.

### C.2 Bayesian logistic regression - Wisconsin cancer data

First, we examine JALA-EM's performance, in the context of training a Bayesian logistic regression model, relative to that of the Particle Gradient Descent (PGD) and Stochastic Optimisation via Unadjusted Langevin (SOUL) algorithms, as introduced in Kuntz et al. (2023) and De Bortoli et al. (2021) respectively. As is the case in the former paper, we utilise the setup described in the latter paper, which we describe in detail below.

**Setup.** The empirical Bayesian logistic regression problem is formulated through considering the Wisconsin Breast Cancer dataset Wolberg and Mangasarian (1990), which can be freely accessed at

https://archive.ics.uci.edu/ml/datasets/breast-cancer+wisconsin+(original).

In particular, this dataset contains 699 samples, obtained between January 1989 and November 1991, where each sample consists of $d_x = 9$ latent variables extracted from digitised images of fine needle aspirates (FNA) of breast masses, such as the *Clump Thickness* and *Marginal Adhesion*, as well as the corresponding malignant or benign diagnosis. Since missing values exist within this dataset, we opted to discard the corresponding samples, resulting in a dataset of $d_y = 683$ samples.

Regarding further data pre-processing, the latent variables, or features, were collated into a matrix $\tilde{X} \in \mathbb{R}^{d_y \times d_x}$, and were then standardised column-wise to have zero mean and unit standard deviation. To be clear, if $\mu_j$ and $s_j$ are the mean and standard deviation of the $j$-th feature column respectively, the normalised features, $X \in \mathbb{R}^{d_y \times d_x}$, are given by $X_{\cdot,j} = (\tilde{X}_{\cdot,j} - \mu_j)/s_j$. Furthermore, note that the response is mapped such that $y \in \{0,1\}^{d_y}$, where $y_i = 0$ represents a benign case and $y_i = 1$ a malignant one. Lastly, note that the pre-processed dataset, $\mathcal{D} = (X, y)$, was then split into a training-validation set, $\mathcal{D}_{train,val}$ and a hold-out test set $\mathcal{D}_{test}$, via an 80-20 stratified split, so that class proportions are maintained across said split.

**Model.** As mentioned above, we adopt an empirical Bayesian logistic regression framework, where, for a single sample $(x_i, y_i)$, in which $x_i \in \mathbb{R}^{d_x}$ and $y_i \in \{0, 1\}$, the corresponding likelihood follows a Bernoulli distribution,

$$p(y_i|x_i, w) = \sigma\left(x_i^\top w\right)^{y_i} \left(1 - \sigma\left(x_i^\top w\right)\right)^{1-y_i},$$

where, to be clear, $w \in \mathbb{R}^{d_x}$ are the regression weights, and $\sigma(z) := (1 + e^{-z})^{-1}$ is the standard logistic function. Following Kuntz et al. (2023), we assign an isotropic Gaussian prior to the regression weights, so that

$$p(w|\theta, \sigma_0^2) = \mathcal{N}\left(w|\theta \cdot \mathbf{1}_{d_x}, \sigma_0^2 I_{d_x}\right),$$

where $\mathbf{1}_{d_x}$ is the $d_x$-dimensional unit vector, and $\sigma_0^2$ is the prior variance. Notably, it is the parameter $\theta \in \mathbb{R}$ that we wish to estimate. Thus, it follows that the unnormalised negative log-posterior, or

the joint potential, for a single particle, indexed by $m$, conditional on the observed data (in the training-validation set) is

$$U(w^{(m)}, \theta | \mathcal{D}_{\text{train,val}}) = \sum_{m=1}^{|\mathcal{D}_{train,val}|} \left( \log \left( 1 + e^{x_i^\top w^{(m)}} \right) - y_i \left( x_i^\top w^{(m)} \right) \right) + \frac{1}{2\sigma_0^2} \left\| w^{(m)} - \theta \mathbf{1}_{d_x} \right\|_2^2$$

where we refer to Appendix D.1.1 for full details. Indeed, as outlined in Proposition 1 of Kuntz et al. (2023), the marginal likelihood has a unique maximiser.

**Approach.** To facilitate a robust comparison of JALA-EM with PGD and SOUL, we implement a systematic and reproducible procedure for model fitting and hyperparameter selection. A critical aspect is the choice of step-sizes, as these significantly influence the dynamics and performance of each algorithm. Notably, JALA-EM and SOUL permit distinct step-sizes for their particle (Langevin) updates and their parameter ($\theta$) updates. This flexibility is not present in PGD, where a single step-size controls both processes. The practical importance of tuning step-sizes is underscored by the toy hierarchical model in Kuntz et al. (2023), where mean-field analysis yields different optimal step-sizes for PGD and its variants presented within the paper.

Therefore, rather than using a fixed global step-size for all algorithms, or all dataset splits, we perform $K$-fold cross-validation on $\mathcal{D}_{train,val}$ to select the step-sizes. Specifically, this tuning is conducted over a predefined grid for the particle update step-sizes and, where applicable (i.e. for SOUL and JALA-EM), for the $\theta$-update step-sizes. The evaluation metric utilised for this hyperparameter tuning, within the cross-validation folds, is the Log Pointwise Predictive Density (LPPD), which assesses the model's average predictive accuracy on unseen data.

Let $\mathcal{P} = \{w^{(k)}\}_{k=1}^N$ be the ensemble of $N$ particles (regression weight vectors) obtained from an algorithm run, to fit the statistical model described above. For each data point $(x_i, y_i) \in \mathcal{D}_{val}$, we first compute the predictive probability of the true observed label, $y_i$, averaged over the $N$ particles,

$$P_{avg}(y_i | x_i, \mathcal{P}) = \frac{1}{N} \sum_{k=1}^N \left[ \sigma(x_i^\top w^{(k)})^{y_i} \left( 1 - \sigma(x_i^\top w^{(k)}) \right)^{1-y_i} \right].$$

The LPPD for $\mathcal{D}_{val}$ is then the average of the natural logarithm of these pointwise predictive probabilities, across the respective dataset,

$$\text{LPPD}(\mathcal{D}_{val}) := \frac{1}{|\mathcal{D}_{val}|} \sum_{i=1}^{|\mathcal{D}_{val}|} \log P_{avg}(y_i | x_i, \mathcal{P}),$$

where a higher LPPD value signifies superior predictive performance of the fitted statistical model, since the mean KL divergence between our classifier and the optimal classifier will be smaller (Kuntz et al., 2023). Notably this quantity is equivalent to the negative of the predictive Negative Log-Likelihood (NLL), which we instead choose to minimise in our implementation. Having selected step-sizes, for the respective algorithms, from our grid, we then re-run the algorithms on the entirety of $\mathcal{D}_{train,val}$, which are then ready to be evaluated on $\mathcal{D}_{test}$.

**Implementation Details.** As is the case in both Akyildiz et al. (2025) and Kuntz et al. (2023), we set the prior variance to $\sigma_0^2 = 5.0$, and also initialise the scalar parameter (to estimate) at $\theta_0 = 0$ for all algorithms. The $N$ particles were initialised by drawing samples independently from the Gaussian prior distribution, $p(w | \theta_0, \sigma_0^2)$, which now notably simplifies to $\mathcal{N}(0, 5I_{d_x})$. Regarding the step-size tuning procedure, we opt for $K = 3$, motivated by the computational cost of the SOUL algorithm, scaling significantly as $N$ increases. We note that early stopping during the tuning occurs if the validation LPPD does not improve by at least $\epsilon_{LPPD} = 1 \times 10^{-5}$ over 10 consecutive evaluations, that are each spaced 10 iterations apart, as we argue that in such cases the stationary phase has been reached. Furthermore, the maximum number of iterations for tuning, during each fold, was set to $K = 500$. To be clear, in the case of SOUL, this refers to the number of outer steps, whereas the number of inner steps is determined by $N$. Also, note that for JALA-EM, we choose $C = 1/1.05$ and utilise systematic resampling in cases in which this threshold is breached, as recommended in Carbone et al. (2023).

To form our grid of step-sizes, relevant to the particle updates, we compute a baseline step-size, which we denote $h_{euler}$. Specifically, $h_{euler}$ is computed to provide a theoretically grounded scale reflecting

the curvature of the energy landscape, and to this end we compute the worst-case upper bound on the Hessian, $H_{bound}$, to obtain a single, globally relevant Lipschitz constant, $L$, for determining $h_{euler}$. Indeed, this is a constant matrix that is larger, in the positive semi-definite sense, than any Hessian encountered, and thus its largest eigenvalue provides an upper bound on $L$ valid across the entire parameter space. In fact, for the Bayesian logistic regression model, this bound takes the form

$$H_{bound} = \frac{1}{4} X_{train,val}^{\top} X_{train,val} + \frac{1}{\sigma_0^2} I_{d_x},$$

where the largest eigenvalue is estimated numerically through the power iteration method Roch (2025), giving our global estimate of $L$. Lastly, to incorporate a small margin of safety, we take $h_{euler} = 0.99/L$. The grid for the particle update step-size was then constructed as 10 linearly spaced values in the range $[0.2 \times h_{euler}, 2.0 \times h_{euler}]$, whereas we limit the grid of $\theta$-update step-sizes to be simply $\{0.05, 0.1, 0.15\}$, again due to the significant time it takes to tune SOUL as $N$ becomes large.

Having selected a step-size, each algorithm was run once on the entire of $\mathcal{D}_{train,val}$, using said step-sizes, for which trajectories were limited to a wall-clock duration of 5.0 seconds, with an iteration milestone of 200 steps, that we indicate, if reached, in the respective visualisations. To be clear, we did not run the algorithms here for the same number of total iterations, but instead the same wall-clock duration.

**Results.** Here, we present the comparative performance of JALA-EM against PGD and SOUL for the Bayesian logistic regression task described above, for the cases that $N = 100$, $N = 50$, and $N = 10$, which is explicitly visualised within Figures 4, 5, and 6 respectively. In particular, we focus on the convergence of the estimates of the global parameter $\theta$, the Kernel Density Estimates (KDEs) for the representative regression weight, $w_2$.

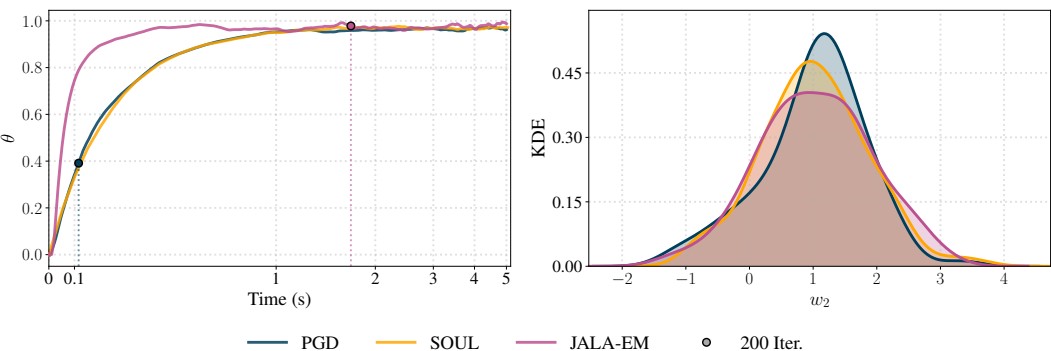

Figure 4: **Bayesian logistic regression:** Parameter estimates for PGD, SOUL, and JALA-EM, for $N = 100$ and tuned step-sizes, with a iteration milestone of 200 indicated (left). KDE of the second coordinate of the posterior approximation for the model weights (right).

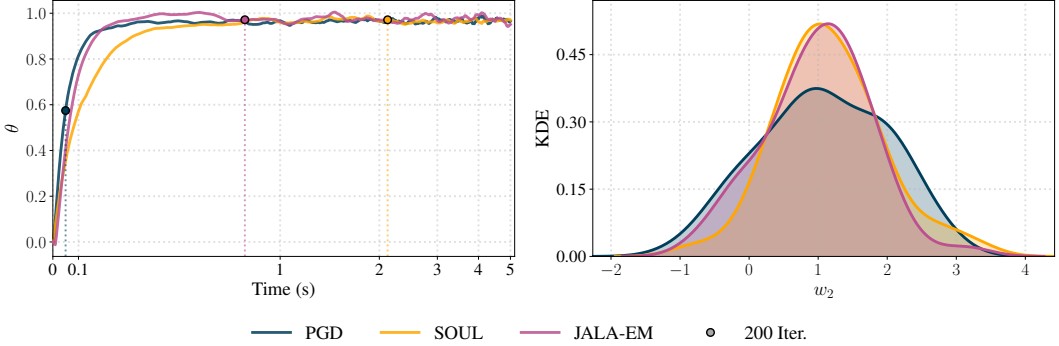

Figure 5: **Bayesian logistic regression:** Parameter estimates for PGD, SOUL, and JALA-EM, for $N = 50$ and tuned step-sizes, with a iteration milestone of 200 indicated (left). KDE of the second coordinate of the posterior approximation for the model weights (right).

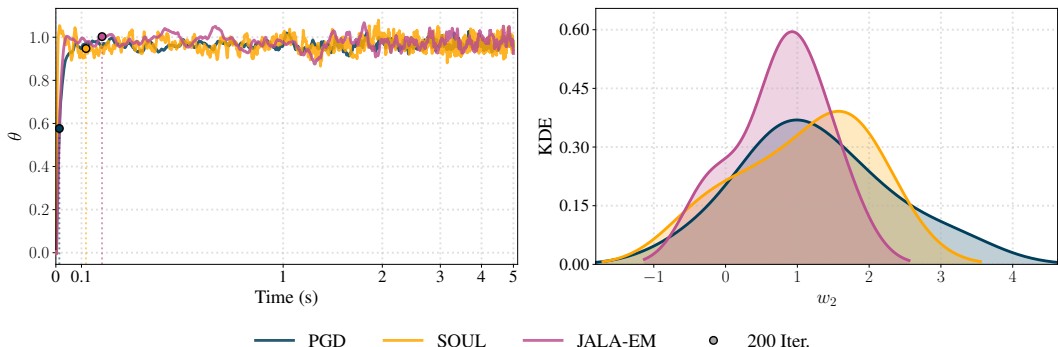

Figure 6: **Bayesian logistic regression:** Parameter estimates for PGD, SOUL, and JALA-EM, for $N = 10$ and tuned step-sizes, with a iteration milestone of 200 indicated (left). KDE of the second coordinate of the posterior approximation for the model weights (right).

Across the particle counts, JALA-EM demonstrates rapid convergence for the parameter $\theta$, consistently matching or even outperforming PGD and SOUL in terms of wall-clock time, while reaching the same estimated value. Although across algorithms all posterior approximations, for the representative regression weight $w_2$, are comparable at $N = 100$, the posterior fidelity of PGD and SOUL noticeably degrades as $N$ decreases. In contrast, JALA-EM maintains high-quality approximations, even with fewer particles, which we attribute to its effective particle management due to reweighting. Although JALA-EM requires an additional step-size to be tuned compared to PGD, it provides a compelling combination of fast parameter estimation and accurate posterior approximation in this setting.

### C.3 Bayesian neural network

To investigate the performance of JALA-EM in a more challenging setting, we opt to apply it to the task of training a Bayesian neural network (BNN). This setting is generally more challenging due to the higher dimensionality of the parameter space and the commonly complex, multimodal nature of the posterior distribution over network weights. Again, we compare JALA-EM with PGD and SOUL.

**Setup.** To begin, we focus on the setup of the binary classification problem, since the multi-class setting is a natural extension. To be clear, using the MNIST handwritten digit dataset, the task is to distinguish between images of the digits 4 and 9, which we remap to classes 0 and 1 respectively. Initially, we filter the dataset for these two digits, and then in the interest of computational speed, as done in Kuntz et al. (2023), we draw a subsample of 1000 images, $f_i \in \mathbb{R}^{D_x = 28 \times 28 = 784}$, and their corresponding labels $l_i \in \{0, 1\}$, ensuring class balance is maintained through stratified sampling. This 1000-sample dataset, $\mathcal{D}_{\text{sub}}$, forms the basis for further processing. Then, image features are standardised by subtracting the pixel-wise mean and dividing by the pixel-wise standard deviation, with statistics computed over $\mathcal{D}_{\text{sub}}$. Subsequently, the processed $\mathcal{D}_{\text{sub}}$ is split into a final training set, $\mathcal{D}_{\text{train}}$, and a hold-out test set, $\mathcal{D}_{\text{test}}$, using an 80-20 stratified split to preserve class proportions.

In the multi-class setting, we instead aim to distinguish between images of the digits 2, 4, 7, and 9, which we remap to the classes 0, 1, 2, and 3 respectively. Again, the dataset is initially filtered for these digits, from which we then draw a subsample of 2500 images through stratified sampling, forming $\mathcal{D}_{\text{sub}}$ here. Subsequently, $\mathcal{D}_{\text{train}}$ and $\mathcal{D}_{\text{test}}$ are formed in the exact same manner as described in the binary setting above.

**Model.** We employ a two-layer fully connected Bayesian neural network, with a notable simplification in the architecture in that the bias parameters are set to zero in both layers. This model structure, including the absence of biases and the empirical Bayes treatment of prior variance hyperparameters (see below), aligns with Kuntz et al. (2023) and Yao et al. (2022).

Specifically, the first layer maps the input feature vector $f_i \in \mathbb{R}^{D_x}$ to a hidden representation using weights $W_1 \in \mathbb{R}^{D_h \times D_x}$ and a hyperbolic tangent activation function, with $D_h = 40$ hidden units. The second layer then maps the hidden representation to the output logits $z_i \in \mathbb{R}^{D_o}$, using weights $W_2 \in \mathbb{R}^{D_o \times D_h}$, where $D_o = 2$ for binary classification and $D_o = 4$ in the multi-class setting. With this in mind, the logits for the $c$-th class of the $i$-th sample are then $z_{ic} = (W_2 \tanh(W_1 f_i))_c$,

while the likelihood for a single data point, $(f_i, l_i)$, is $p(l_i|f_i, W_1, W_2) = (\text{softmax}(z_i))_{l_i}$. Note this is equivalent to using a categorical cross-entropy loss where the log-likelihood term is $(\log \text{softmax}(z_i))_{l_i}$.

Now, isotropic Gaussian priors are then assigned to the weights of both layers, so that $p(W_1|\alpha) = \mathcal{N}(W_1|\mathbf{0}, e^{2\alpha}I_{D_h \times D_x})$ and $p(W_2|\beta) = \mathcal{N}(W_2|\mathbf{0}, e^{2\beta}I_{D_o \times D_h})$. The parameters $\alpha \in \mathbb{R}$ and $\beta \in \mathbb{R}$ control the logarithm of the standard deviation of these priors and are themselves estimated from the data as part of an empirical Bayes procedure, collectively denoted $\theta = (\alpha, \beta)$. Let $D_{W_1} = D_h D_x$ and $D_{W_2} = D_o D_h$ be the total number of parameters in $W_1$ and $W_2$, respectively. Then, the joint probability density of the weights and the training labels, $\mathcal{L}_{\text{train}} = \{(f_i, l_i)\}_{i=1}^{|\mathcal{D}_{\text{train}}|}$, given $\theta$, is:

$$p(W_1, W_2, \mathcal{L}_{\text{train}}|\theta) = p(W_1|\alpha)p(W_2|\beta) \prod_{(f_i,l_i) \in \mathcal{L}_{\text{train}}} p(l_i|f_i, W_1, W_2).$$

The unnormalised negative log-posterior, relevant for the Langevin-based algorithms, is then $U(W_1, W_2, \theta|\mathcal{L}_{\text{train}}) = -\log p(W_1, W_2, \mathcal{L}_{\text{train}}|\theta)$, which implicitly includes the log-prior terms.

**Approach.** The primary goal is to compare the efficacy of JALA-EM against PGD and SOUL in training this BNN. Both the LPPD and classification Test Error on $\mathcal{D}_{\text{test}}$ are used to evaluate performance. Indeed, given an ensemble of $N$ particles $\{(W_1^{(k)}, W_2^{(k)})\}_{k=1}^N$, the average predictive probability for the true label $l_i$ of a test sample $f_i$ is

$$P_{\text{avg}}(l_i|f_i, \{(W_1^{(k)}, W_2^{(k)})\}_{k=1}^N) = \frac{1}{N} \sum_{k=1}^N \left[ \text{softmax}(W_2^{(k)} \tanh(W_1^{(k)} f_i)) \right]_{l_i}.$$

The LPPD for $\mathcal{D}_{\text{test}}$ is then:

$$\text{LPPD}(\mathcal{D}_{\text{test}}) := \frac{1}{|\mathcal{D}_{\text{test}}|} \sum_{(f_i,l_i) \in \mathcal{D}_{\text{test}}} \log(\max(10^{-30}, P_{\text{avg}}(l_i|f_i, \{(W_1^{(k)}, W_2^{(k)})\}_{k=1}^N))),$$

where probabilities are clipped here in the interest of numerical stability.

**Implementation Details.** First, we describe the details for the binary classification setting. Regarding initialisation, the initial parameter values are $\alpha_0 = 0.0$ and $\beta_0 = 0.0$. For all algorithms, $N = 100$ particles are initialised by drawing weights from their respective priors $p(W_1|\alpha_0)$ and $p(W_2|\beta_0)$, which simplifies to $\mathcal{N}(\mathbf{0}, I)$ given $\alpha_0 = \beta_0 = 0$. Furthermore, each algorithm runs for $K = K_{\text{common\_iters}} = 500$ iterations. In contrast to the Bayesian logistic regression experiment (see Appendix C.2), this experiment utilises a global, fixed step-size of $0.1$, for all algorithms, rather than tuning them via $K$-fold cross-validation, as this is an example where computational (or expertise) limitations prohibit comprehensive fine-tuning. As such, all algorithms, including JALA-EM, are implemented in JAX. Note that for JALA-EM, we choose $C = 1/1.05$ as was the case before. We also note that, in the interest of numerical stability some extra clipping occurs for significantly large norms, and the gradients used for updates are normalised as in Kuntz et al. (2023). The most notable change in the multi-class setting is that we instead utilise $N = 50$ particles, for all algorithms, in the interest of computational speed.

**Results.** In the binary classification scenario, we observe short transient phases in the evolution of parameter estimates—albeit converging to different local maxima—as well as in evaluation metrics for PGD, SOUL, and JALA-EM, as illustrated in Figure 3. Notably, the predictive performance of JALA-EM is comparable to that of SOUL, which is to be expected due to the correlations that exist between particles due to the resampling mechanism present in JALA-EM. Although PGD converges faster in terms of wall-clock time, it does not support marginal likelihood estimation, which is a key advantage of our method. In the multi-class scenario we obtain average final misclassification error rates (and standard deviations) of 4.74% (0.34%), 9.24% (0.93%), and 9.20% (1.27%) for PGD, JALA-EM, and SOUL respectively, across 10 repeats. Indeed, one can vary the number of hidden neurons, and to this end we repeated the experiment for $D_h = 512$ hidden neurons, resulting in a BNN of 403,456 latent variables, where note we had BNNs of 31,440 and 31,520 latent variables previously, for the binary and multi-class scenarios respectively. In this higher-dimensional case, the average final misclassification error rates (and standard deviations) obtained were 4.98% (0.42%), 7.20% (0.81%), and 8.02% (1.02%) for PGD, JALA-EM, and SOUL respectively, across 10 repeats.

Overall, this example highlights the reasonable scalability of our proposal to more complex and high-dimensional inference problems.

## C.4 Error model selection - Bayesian regression

Having investigated JALA-EM's efficacy in parameter estimation within both a straightforward and more complex task, we now examine the algorithm's capability to estimate marginal likelihoods. In fact, this is a dual capability since parameters of interest are still estimated. In particular, we consider the distinct, yet related challenge of Bayesian model selection.

**Setup.** We consider a model selection problem for a dataset, $\mathcal{D} = \{(x_i, y_i)\}_{i=1}^{d_y}$, generated by an underlying model, $\mathcal{G}$, where $x_i \in \mathbb{R}^{d_x}$ and $y_i \in \mathbb{R}$, which we collate into $X \in \mathbb{R}^{d_y \times d_x}$ and $y \in \mathbb{R}^{d_y}$ respectively. Specifically, we focus on Bayesian linear regression models for $\mathcal{D}$, that aim to capture the relationship $y_i = X_i w + \varepsilon_i$ through a latent weight vector $w \in \mathbb{R}^{d_x}$, where $\varepsilon_i$ represents the observation error. Indeed, JALA-EM can not only be leveraged to estimate the parameters of a given model, but also can be used to estimate the marginal likelihood, facilitating model selection through the use of, say, a Bayes Factor.

**Models.** In this experiment, we consider a pair of competing, nested Bayesian linear regression models, where for both models, an isotropic Gaussian prior is placed on these weights, $p(w|\alpha) = \mathcal{N}(0, \alpha^{-1} I_{d_x})$, where $\alpha > 0$ is the *precision*, however, the models importantly differ in their assumptions about $\varepsilon_i$.

The first model we consider, namely the Gaussian regression model, $\mathcal{M}_G$, assumes independent and identically distributed (i.i.d.) Gaussian errors, $\varepsilon_i \sim \mathcal{N}(0, \sigma^2)$, leading to the likelihood,

$$p(y|X, w, \sigma^2, \mathcal{M}_G) = \prod_{i=1}^{d_y} \mathcal{N}(y_i|X_i w, \sigma^2) = \mathcal{N}(y|Xw, \sigma^2 I_{d_y}),$$

where, for numerical stability, and to ensure the positivity of both $\sigma^2$ and $\alpha$, we choose to work with the logarithmic parametrisations of the parameters of interest. To be clear, we estimate $\theta_G = (\phi_1, \phi_2)$, with $\phi_1 = \log \sigma^2$ and $\phi_2 = \log \alpha$.

The second model we consider is the Student-t regression model, $\mathcal{M}_T$, which assumes i.i.d. Student-t distributed errors, $\varepsilon_i \sim \text{Student-t}(0, \sigma^2, \nu)$, leading to greater robustness to outliers, and results in the likelihood

$$p(y|X, w, \sigma^2, \nu, \mathcal{M}_T) = \prod_{i=1}^{d_y} \frac{\Gamma((\nu+1)/2)}{\Gamma(\nu/2)\sqrt{\pi \nu \sigma^2}} \left( 1 + \frac{(y_i - X_i w)^2}{\nu \sigma^2} \right)^{-(\nu+1)/2},$$

where the additional parameter, $\nu$, is the *degrees of freedom*. Here, an exponential prior is placed on $\nu$, that is $p(\nu|\lambda_\nu) = \lambda_\nu e^{-\lambda_\nu \nu}$, with the rate parameter, $\lambda_\nu = 0.1$ fixed throughout all experiments. Again, for numerical stability, and to ensure positivity, we work with logarithmic parametrisations, and so estimate $\theta_T = (\phi_1, \phi_2, \phi_3)$, with $\phi_3 = \log \nu$, while $\phi_1$ and $\phi_2$ are as before. Notably, as $\nu \to \infty$, the Student-t distribution approaches the Gaussian distribution, rendering $\mathcal{M}_G$ nested within $\mathcal{M}_T$. To ensure that $\mathcal{M}_T$ provides distinct heavy-tailed modelling and to avoid identifiability issues with $\mathcal{M}_G$, particularly when $\mathcal{G} = \mathcal{M}_G$, we constrain $\nu \in [0.2, 5.0]$, achieved in our implementation by clipping $\phi_3 \in [\log(0.2), \log(5.0)]$.

**Approach.** For the synthetic dataset, $\mathcal{D}$, generated by the regression model $\mathcal{G} \in \{\mathcal{M}_G, \mathcal{M}_T\}$, we employ JALA-EM to iteratively refine our initial estimates for $\theta_G$ and $\theta_T$, while concurrently estimating the corresponding model's marginal likelihood. Indeed, after $K$ iterations, by Proposition 1, we have that our estimate for the log marginal likelihood is given by

$$\log \hat{Z}_{\mathcal{M}, K} = \log Z_{\mathcal{M}, 0} + \log \left( \sum_{i=1}^{N} e^{A_K^i} \right) - \log N \approx \log p(y|X, \theta_{\mathcal{M}, K}, \mathcal{M}),$$

where $Z_{\mathcal{M}, 0}$ is the marginal likelihood evaluated at the initial estimates, $\theta_{\mathcal{M}, 0}$. We adopt the following version of (14) which includes resampling as usual in SMC contexts, cfr. Del Moral et al. (2006):

$$\log \hat{Z}_{\mathcal{M}, K} = \log Z_{\mathcal{M}, 0} + \sum_{j=1}^{J} \log \left( \frac{1}{N} \sum_{i=1}^{N} \exp \left( \sum_{m=k_{j-1}+1}^{k_j} A_m^i \right) \right), \quad k_0 = 1 \tag{36}$$

where $k_j$ is the step at which the $j$-th resampling event occurs, with $j = 1, \ldots, J$, whilst $A_m^i$ is the log-incremental weight of particle $i$ at step $m$, and $\sum_{m=k_{j-1}+1}^{k_j} A_m^i$ represents the cumulative log-weight of particle $i$ between two resampling steps.

Notably, the method for computing $Z_{\mathcal{M},0}$ is model dependent, and for $\mathcal{M}_G$ this is analytically tractable,

$$\log Z_{\mathcal{M}_G,0} = \log \mathcal{N}(y|0, \sigma^2 I_{d_y} + \alpha_0^{-1} XX^\top) = -\frac{1}{2}(d_y \log(2\pi) + \log(\det(\Sigma_0)) + y^\top \Sigma_0^{-1} y),$$

where $\Sigma_0 = \sigma_0^2 I_{d_y} + \alpha_0^{-1} XX^\top$ (see Appendix D.2.1). To be clear, $\sigma_0^2$ and $\alpha_0$ form the initial parameter estimate, $\theta_{G,0}$.

For $\mathcal{M}_T$, however, $Z_{\mathcal{M}_T,0}$ is generally intractable, and so we estimate it using Importance Sampling (IS), drawing $S$ samples, $\{w_s\}_{s=1}^S$, from a proposal distribution $q(w)$. Specifically, we choose $q(w) = \mathcal{N}(w|\mu_q, \Sigma_q)$, where $\mu_q$ and $\Sigma_q$ are the posterior mean and covariance, respectively, of $w$ under a Gaussian likelihood model, $p(w|X, y, \sigma_0^2, \alpha_0)p(w|\alpha_0)$, using the initial $\sigma_0^2$ and $\alpha_0$ from $\theta_{T,0}$. The IS estimate of the initial marginal likelihood is thus

$$\hat{Z}_{\mathcal{M}_T,0} \approx \frac{1}{S} \sum_{s=1}^S \frac{p(y|X, w_s, \sigma_0^2, \nu_0, \mathcal{M}_T)p(w_s|\alpha_0)}{q(w_s)},$$

where $\nu_0$ also makes up $\theta_{T,0}$.

It is then natural to perform model selection using the logarithm of the approximate Bayes Factor, given by $\log(BF) \approx \log \hat{Z}_{\mathcal{M}_G,K} - \log \hat{Z}_{\mathcal{M}_T,K}$, where a positive value indicates, conditioned on the estimated $\theta_G$ and $\theta_T$, that the data provides greater evidence for $\mathcal{M}_G$ as the true data generating process for $\mathcal{D}$.

**Implementation Details.** In our numerical experiments, synthetic datasets were generated, comprising of $d_y = 500$ samples and $d_x = 8$ features, where to be clear, the feature vectors $x_i$, for each sample, were drawn i.i.d. from a standard multivariate Gaussian, $\mathcal{N}(0, I_{d_x})$. Indeed, two distinct scenarios for the true generating process, $\mathcal{G}$, are considered, where in the first $\mathcal{G} = \mathcal{M}_G$, and observation errors were drawn from $\varepsilon_i \sim \mathcal{N}(0, \sigma_*^2)$, whereas in the Student-t case, $\mathcal{G} = \mathcal{M}_T$, observation errors were drawn from $\varepsilon_i \sim$ Student-t$(0, \sigma_*^2, \nu_*)$. In both cases, the true weight precision is $\alpha_* = 1.0$, as is the true error variance $\sigma_* = 1.0$, while the true degrees of freedom is set to $\nu_* = 4.0$. For each of the data-generating scenarios, we note that the complete experimental trial, that is the data generation, fitting of $\mathcal{M}_G$ and $\mathcal{M}_T$ via JALA-EM, and subsequent model selection, was repeated 100 times, for which we report the proportion of trials in which the correct model was recovered. To be clear, to generate a single $\mathcal{D}$, a single true weight vector, $w_*$, is drawn from $\mathcal{N}(0, \alpha_*^{-1} I_{d_x}) = \mathcal{N}(0, I_{d_x})$. Indeed, we utilise different random seeds for each experimental trial to ensure variability in both the synthetic dataset generation and in the stochastic elements of JALA-EM.

Regarding the configuration of JALA-EM, we initialise model parameters perturbed from their true values, so that $\theta_{\mathcal{M},0} = (\log \sigma_*^2 + 1, \log \alpha_* + 1, \dots) = (1, 1, \dots)$, where we set $\log \nu_0 = \log \nu_* + 1 = \log(4.0) + 1$ when $\mathcal{G} = \mathcal{M}_T$, and $\log \nu_0 = \log(5.0)$ when $\mathcal{G} = \mathcal{M}_G$, corresponding to the upper limit of our constraint for $\nu$. The algorithm is run for $K = 250$ iterations, using $N = 50$ particles, with a Langevin dynamic step-size of $h = 5 \times 10^{-5}$, while the parameter optimisation learning rate is $\eta = 5 \times 10^{-3}$, where OPT is in fact Adam (Kingma and Ba, 2015), with $\beta_1 = 0.9$, to demonstrate optimisers other than SGD can be leveraged. In the case of $\mathcal{M}_G$, particles are drawn directly from the analytically available posterior, $p(w|X, y, \theta_{G,0})$, which is notably a Gaussian. For $\mathcal{M}_T$, however, particles are generated by $N$ independent short MCMC runs, where each chain targets the posterior $p(w|X, y, \theta_{T,0})$, and is initialised from the prior $p(w|\alpha_0)$. Specifically, each MCMC run generates a particle by evolving a chain for 200 steps, using Unadjusted Langevin Algorithm (ULA) updates, so that its final state is taken as the particle sample. The ULA step size at step $t$, denoted $\epsilon_t$, is adapted dynamically based on the L2-norm of the gradient of log-target density, $g_t = \|\nabla_w \log p(w_t|\cdot)\|$. In fact, we initialise $\epsilon_0 = 1 \times 10^{-3}$ and let $\epsilon_{t+1} = 0.9\epsilon_t$ if $g_t > 1000 d_x$ and $\epsilon_t > 10^{-6}$. Conversely, if $g_t < 10 d_x$ and $\epsilon_t < 0.1$ then we let $\epsilon_{t+1} = 1.05\epsilon_t$, and otherwise $\epsilon_t$ is left unchanged. To ensure we are consistent with Proposition 1, we fix the resampling threshold to $C = 0$, so that particle resampling is essentially disabled, however note that one can also track $Z_k$ through the resampling step, as outlined in Section 5. Lastly, we note that during the IS, $S = 5000$ samples are drawn.

**Results.** We begin by commenting on the behaviour of JALA-EM for single representative (at least for the 100 repeats) synthetic datasets, each comprising of $d_y = 500$ samples, as illustrated in Figure 7. Indeed, not only is JALA-EM's ability to converge quickly to sensible parameter estimates illustrated, but so is its capability to differentiate between the true and misspecified model.

In the case where the true underlying model is Gaussian, that is $\mathcal{G} = \mathcal{M}_G$, JALA-EM fitting $\mathcal{M}_G$ quickly estimated the ML-II (see Appendix D.2.2) values of $(\phi_1, \phi_2)$, and what is more, the estimated marginal likelihood quickly stabilises close to its true analytical value. To be clear, by this we mean the analytical value of the marginal likelihood corresponding to the true parameters, $\theta_*$, and we denote this by $\log Z_{\mathcal{M}, \theta_*}$. An important, yet subtly different quantity, is the analytical value of the marginal likelihood corresponding to the parameters estimated at the iteration step $k$, which we denote by $\log Z_{\mathcal{M}, \theta_k}$. As outlined above, this is a tractable quantity for $\mathcal{M}_G$, and we observe this to correspond exactly with the estimates generated by JALA-EM in this case, importantly corroborating Proposition 1.

Conversely, when the true underlying model is the Student-t, that is $\mathcal{G} = \mathcal{M}_T$, JALA-EM fitting $\mathcal{M}_T$ is more effective at recovering the true parameter values, and importantly its $\log \hat{Z}_{\mathcal{M}_T, \theta_k}$ stabilises at a higher value. In fact, in both scenarios the estimated marginal likelihood trajectories differ significantly, suggesting potential for JALA-EM to perform robust Bayesian model selection.

Notably, when applying the previously described procedure with $d_y = 1500$, instead of $d_y = 500$, samples, as illustrated in Figure 8, we observe broadly consistent behaviour, albeit with more decisive model discrimination. In fact, with more data the parameter estimates for the correctly specified models maintain their accuracy, while the separation between estimated marginal likelihoods becomes more distinct. This increased distinction is consistent with Bayesian principles, where the increased amount of data provides stronger evidence, and indicates JALA-EM's ability to effectively leverage larger datasets for more confident model selection.

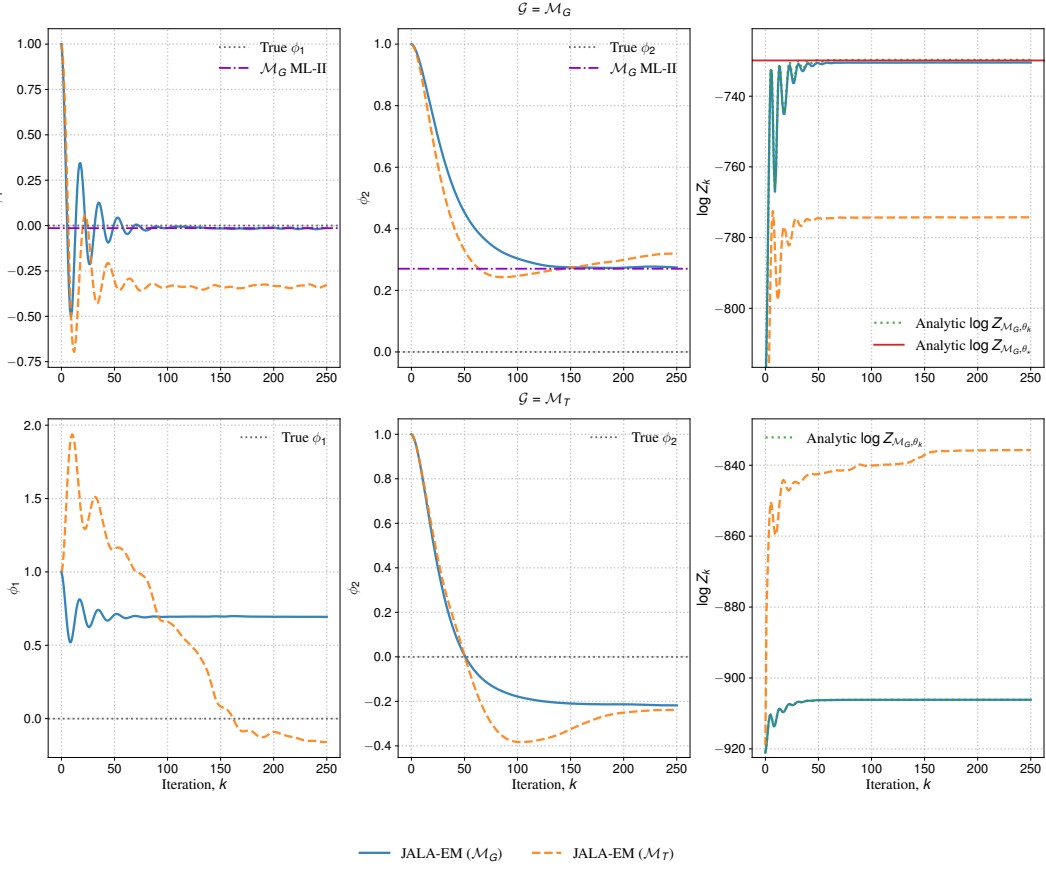

Figure 7: **Error model selection:** Parameter estimates (left & middle) and marginal likelihood estimates (right) for JALA-EM fitting $\mathcal{M}_G$ and $\mathcal{M}_T$, where the true underlying model is the former (top) and the latter (bottom). Here, $d_y = 500$, $d_x = 8$, and $N = 50$. Parameter estimates are initialised at values perturbed away from the true values.

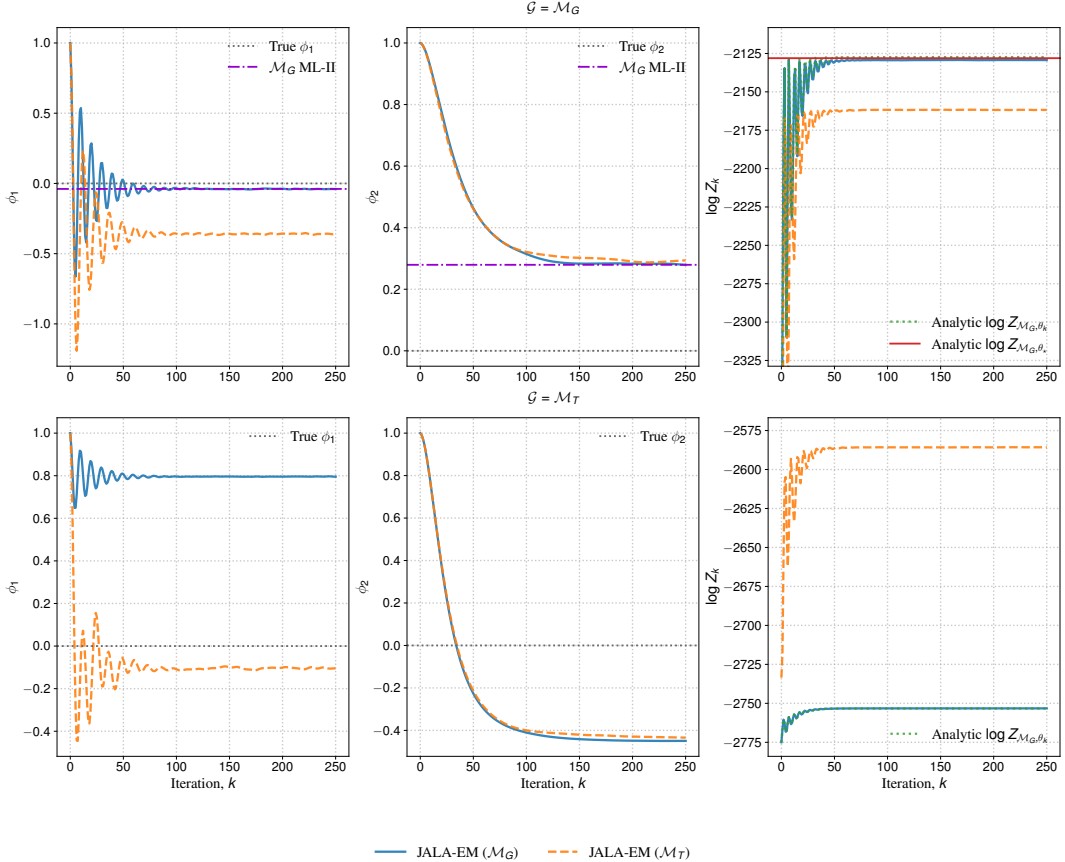

Figure 8: **Error model selection:** Parameter estimates (left & middle) and marginal likelihood estimates (right) for JALA-EM fitting $\mathcal{M}_G$ and $\mathcal{M}_T$, where the true underlying model is the former (top) and the latter (bottom). Here, $d_y = 1500$, $d_x = 8$, and $N = 50$. Parameter estimates are initialised at values perturbed away from the true values.

Regarding the aforementioned 100 repetitions of the core experimental trial, with model selection being performed by selecting the model with the higher $\log \hat{Z}_{\mathcal{M},\theta_K}$, we found that the underlying model class was correctly identified 100% of the time when $\mathcal{G} = \mathcal{M}_G$, and 99% of the time when $\mathcal{G} = \mathcal{M}_T$.

## C.5 Model order selection - Bayesian regression

To further evaluate JALA-EM's performance, in a more complex model selection context, we consider the challenge of Bayesian model selection in a polynomial regression setting. Once again, this highlights JALA-EM's dual capability to estimate both parameters of interest and marginal likelihoods.

**Setup.** In a subtly distinct manner, we now consider a model selection problem for a dataset, $\mathcal{D} = \{(x_i, y_i)\}_{i=1}^{d_y}$, generated by an underlying model, $\mathcal{G}$, where both $x_i \in \mathbb{R}$ and $y_i \in \mathbb{R}$. Specifically, we focus on a family of Bayesian linear regression models that aim to capture the relationship $y_i = \varphi_p (x_i)^\top w_p + \varepsilon_i$ through a latent weight vector $w_p \in \mathbb{R}^{d_p}$, where $\varphi_p (x_i) \in \mathbb{R}^{d_p}$ is a model specific feature vector and $\varepsilon_i$ represents the observation error. To be clear, we then collate $\varphi_p (x_i)$ and $y_i$ into $X_p \in \mathbb{R}^{d_y \times d_p}$ and $y \in \mathbb{R}^{d_y}$ in a similar manner as seen in Appendix C.4.

**Models.** In particular, we consider a larger set of competing, nested Bayesian linear regression models, $\{\mathcal{M}_p\}_{p=1}^{P}$. Notably, these models share the same probabilistic structure for both their weights and observation errors, differing instead in the model specific features utilised.

Once again, for all models, an isotropic Gaussian prior, $p(w|\alpha) = \mathcal{N}(0, \alpha^{-1}I_{d_p})$ is placed on the weights, as are i.i.d Gaussian errors assumed. Indeed, this leads to the likelihood,

$$p(y|X_p, w_p, \sigma^2, \mathcal{M}_p) = \prod_{i=1}^{d_y} \mathcal{N}(y_i|\phi_p(x_i)^\top w_p, \sigma^2) = \mathcal{N}(y|X_p w_p, \sigma^2 I_{d_y}),$$

where we again choose to estimate $\theta_G = (\phi_1, \phi_2)$, with $\phi_1 = \log \sigma^2$ and $\phi_2 = \log \alpha$, for numerical stability and guarantees of positivity.

Indeed, the $p$-th model in the candidate set, $\mathcal{M}_p$, corresponds to a polynomial regression model of order $p$, that assumes a model specific feature vector $\varphi_p(x_i)$ derived from the polynomial basis expansion up to and including order $p$, so that $\varphi_p(x_i) = [1, x_i, \ldots, x_i^p]$. As a consequence, we have that $d_p = p + 1$, and so $\varphi_p(x_i) \in \mathbb{R}^{p+1}$, $w_p \in \mathbb{R}^{p+1}$, and $X_p \in \mathbb{R}^{d_y \times (p+1)}$.

**Approach.** For the synthetic dataset, $\mathcal{D}$, generated by the true regression model $\mathcal{G} = \mathcal{M}_{p_\star}$, we utilise JALA-EM to fit each of the $P$ candidate models, with the aim of estimating $p_\star$. For each candidate, JALA-EM iteratively refines the initial estimates for $\theta$, while concurrently estimating the corresponding model's marginal likelihood. Indeed, after $K$ iterations, by Proposition 1, we have that our estimate, of model $\mathcal{M}_p$, for the log marginal likelihood is given by

$$\log \hat{Z}_{\mathcal{M}_p, K} = \log Z_{\mathcal{M}_p, 0} + \log \left( \sum_{i=1}^N e^{A_K^i} \right) - \log N \approx \log p(y|X_p, \theta_{\mathcal{M}_p, K}, \mathcal{M}_p),$$

where $Z_{\mathcal{M}_p, 0}$ is the marginal likelihood evaluated at the initial estimates, $\theta_{\mathcal{M}_p, 0}$, for model $\mathcal{M}_p$.

Notably, in this case, the method for computing $Z_{\mathcal{M}_p, 0}$ is identical for all candidate models, and since we have Gaussian errors in all cases, it is analytically tractable for all orders,

$$\log Z_{\mathcal{M}_p, 0} = \log \mathcal{N}(y|0, \sigma^2 I_{d_y} + \alpha_0^{-1} X_p X_p^\top) = -\frac{1}{2}(d_y \log(2\pi) + \log(\det(\Sigma_0)) + y^\top \Sigma_0^{-1} y),$$

where $\Sigma_0 = \sigma_0^2 I_{d_y} + \alpha_0^{-1} X_p X_p^\top$ (see Appendix D.2.1). To be clear, $\sigma_0^2$ and $\alpha_0$ form the initial parameter estimate, $\theta_{\mathcal{M}_p, 0}$.

Model selection is thus naturally performed by simply selecting the model order that maximises this quantity, namely $\hat{p} = \arg \max_{p \in [P]} \log \hat{Z}_{\mathcal{M}_p, K}$.

As a competitive baseline, we compare our method to an approach that leverages Ordinary Least Squares (OLS) for weight estimation, followed by selecting the model with the lowest Bayesian Information Criterion (BIC). To be clear, for each candidate model, we first compute the maximum likelihood estimate (MLE) for the weight vector, $\hat{w}_{p,\text{MLE}}$ by minimising the residual sum of squares (RSS). Indeed, under the assumption of Gaussian errors, this reduces to standard OLS, whose solution is given by

$$\hat{w}_{p,\text{MLE}} = (X_p^\top X_p)^{-1} X_p^\top y,$$

where $X_p$ has full rank, so that the Gram matrix is invertible and $\hat{w}_{p,\text{MLE}}$ is established as the desired minimiser of the RSS. The MLE for the observation variance, $\hat{\sigma}_{p,\text{MLE}}$ is computed in a similar manner,

$$\hat{\sigma}_{p,\text{MLE}}^2 = \frac{1}{d_y} \|y - X_p \hat{w}_{p,\text{MLE}}\|_2^2,$$

from which the BIC for $\mathcal{M}_p$ is then calculated,

$$\text{BIC}_p = (p + 2) \log d_y - 2 \log p(y|X_p, \hat{w}_{p,\text{MLE}}, \hat{\sigma}_{p,\text{MLE}}^2, \mathcal{M}_p),$$

where we subsequently select the model order that minimises the BIC, that is we select the model order $\hat{p}_{\text{base}} = \arg \min_{p \in [P]} \text{BIC}_p$.

**Implementation Details.** In the numerical experiments, synthetic datasets were generated, comprising of $d_y$ samples. In this case, the values of the single feature $x_i$, from which the model specific feature vectors are derived, were drawn i.i.d from a Uniform distribution, $\mathcal{U}[-2.5, 2.5]$. Indeed, the observation errors were drawn from $\varepsilon_i \sim \mathcal{N}(0, \sigma_\star^2)$, where the true error variance is $\sigma_\star^2 = 7.5$,

whereas the true weight precision is $\alpha_\star = 1.0$ so that the true weight vector, $w_\ast$, is drawn from $\mathcal{N}(0, \alpha_\ast^{-1} I_{d_p}) = \mathcal{N}(0, I_{p+1})$. Notably, each experimental trial has an associated true polynomial order, $p_\star$, so that $y_i = \varphi_{p_\star}(x_i)^\top w_\star + \varepsilon_i$, where we have $\varphi_{p_\star}(x_i) = [1, x_i, \ldots, x_i^{p_\star}]$.

Regarding the configuration of JALA-EM, we again initialise model parameters perturbed from their true values, so that $\theta_{\mathcal{M}_p,0} = (\log \sigma_\ast^2 + 1, \log \alpha_\ast + 1) = (1, 1)$. The algorithm is run for $K = 200$ iterations, using $N = 50$ particles, with a Langevin dynamic step-size of $h = 1 \times 10^{-6}$, while the parameter optimisation learning rate is $\eta = 5 \times 10^{-3}$, where OPT is again Adam, with $\beta_1 = 0.9$. For all candidate models, which share a Gaussian likelihood structure, particles are drawn directly from the analytically available posterior $p(w_p | X_p, y, \theta_{\mathcal{M}_p,0})$. Once again, to ensure we are consistent with Proposition 1, we fix the resampling threshold to $C = 0$.

**Results.** To evaluate the performance of JALA-EM in a robust and systematic manner, for each of $d_y \in \{100, 250, 500\}$ separately, we iterated through a range of true orders, $p_\star \in [2, 8]$, and ran 100 repeats of the core experimental trial, with different random seeds for each trial. To be clear, the task for both JALA-EM and the aforementioned baseline is to select the true model order from a candidate set of polynomial degrees, $p \in [1, 10]$. In order to reward close estimates, for cases in which the estimated order is incorrect, we choose to utilise the Mean Absolute Error (MAE), over Accuracy, for evaluation. The results, across the true orders, can be found in Table 1.

Indeed, we observe JALA-EM to match or outperform the baseline for all model orders considered, achieving notably lower average MAE in cases where the underlying polynomial is of higher-order. In such cases, the true model is more complex, possessing a higher-dimensional latent variable space, in which the Bayesian treatment of JALA-EM appears more effective at balancing model fit and complexity. In contrast, the baseline's approach, which notably relies on a point estimate, $\hat{w}_{p,\text{MLE}}$, and the BIC approximation, is less reliable, and in fact exhibits a propensity to underfit, due to the marginal improvement in log-likelihood being outweighed by increases in the complexity penalty for our non-asymptotic regimes.

Table 1: Average MAE values of model order estimates, for true orders $p_\star \in [2, 8]$, obtained over 100 experimental trials, for JALA-EM and the OLS & BIC baseline. Here, $d_y \in \{100, 250, 500\}$ and $N = 50$, whilst the candidate model orders are $p \in [1, 10]$. Parameter estimates are initialised at values perturbed away from their true values.

| True Order | JALA-EM (MAE) | | | OLS & BIC Baseline (MAE) | | |
|---|---|---|---|---|---|---|
| | $d_y = 100$ | $d_y = 250$ | $d_y = 500$ | $d_y = 100$ | $d_y = 250$ | $d_y = 500$ |
| 2 | 0.67 | 0.45 | 0.35 | 0.64 | 0.46 | 0.35 |
| 3 | 0.73 | 0.41 | 0.25 | 0.90 | 0.42 | 0.34 |
| 4 | 0.73 | 0.49 | 0.30 | 0.80 | 0.60 | 0.30 |
| 5 | 0.51 | 0.40 | 0.27 | 0.64 | 0.42 | 0.35 |
| 6 | 0.35 | 0.31 | 0.18 | 0.49 | 0.32 | 0.23 |
| 7 | 0.23 | 0.15 | 0.09 | 0.44 | 0.28 | 0.15 |
| 8 | 0.28 | 0.19 | 0.13 | 0.37 | 0.30 | 0.20 |

# D  Further theoretical results for experiments

## D.1  Bayesian logistic regression - Wisconsin cancer data

### D.1.1  Derivation of the negative log-posterior

Here, we detail the derivation of the unnormalised, negative log-posterior, for a single particle $w^{(k)}$, conditional on the observed data $\mathcal{D}_{\text{train, val}} = \{(x_i, y_i)\}_{i=1}^{|\mathcal{D}_{\text{train, val}}|}$ and the parameter $\theta$, which we denote $U(w^{(k)}, \theta | \mathcal{D}_{\text{train, val}})$.

Indeed, the posterior of the weights $w$ and parameter $\theta$, given the data $\mathcal{D}_{\text{train, val}}$, follows from Bayes' theorem,

$$p(w, \theta | \mathcal{D}_{\text{train, val}}) \propto p(\mathcal{D}_{\text{train, val}} | w) p(w | \theta, \sigma_0^2) p(\theta).$$

For a fixed $\theta$, as is the case when evaluating the potential for a specific particle $w^{(k)}$, and by treating the prior $p(\theta)$ as uniform, the unnormalised posterior for $w^{(k)}$ is

$$p(w^{(k)}|\mathcal{D}_{\text{train, val}}, \theta, \sigma_0^2) \propto p(\mathcal{D}_{\text{train, val}}|w^{(k)})p(w^{(k)}|\theta, \sigma_0^2).$$

The potential function $U(w^{(k)}, \theta|\mathcal{D}_{\text{train, val}})$ is defined as the negative logarithm of this quantity, albeit up to an additive constant $c \in \mathbb{R}$

$$U(w^{(k)}, \theta|\mathcal{D}_{\text{train, val}}) = -\log p(\mathcal{D}_{\text{train, val}}|w^{(k)}) - \log p(w^{(k)}|\theta, \sigma_0^2) + c,$$

for which we derive the forms of each term below.

To begin, note that the negative log-likelihood for $\mathcal{D}_{\text{train, val}}$ is

$$-\log p(\mathcal{D}_{\text{train, val}}|w^{(k)}) = -\sum_{i=1}^{|\mathcal{D}_{\text{train, val}}|} \log p(y_i|x_i, w^{(k)})$$

$$= \sum_{i=1}^{|\mathcal{D}_{\text{train, val}}|} \left[ \log(1 + e^{x_i^\top w^{(k)}}) - y_i x_i^\top w^{(k)} \right].$$

Now, recall that the prior for the weights $w^{(k)}$, given $\theta$ and $\sigma_0^2$, is chosen as an isotropic Gaussian,

$$p(w^{(k)}|\theta, \sigma_0^2) = \mathcal{N}(w^{(k)}|\theta \cdot \mathbf{1}_{d_x}, \sigma_0^2 I_{d_x}).$$

Thus we have that

$$p(w^{(k)}|\theta, \sigma_0^2) = \frac{1}{(2\pi\sigma_0^2)^{d_x/2}} \exp\left(-\frac{1}{2\sigma_0^2}\|w^{(k)} - \theta \cdot \mathbf{1}_{d_x}\|_2^2\right),$$

and so we obtain

$$-\log p(w^{(k)}|\theta, \sigma_0^2) = -\log\left(\frac{1}{(2\pi\sigma_0^2)^{d_x/2}} \exp\left(-\frac{1}{2\sigma_0^2}\|w^{(k)} - \theta \cdot \mathbf{1}_{d_x}\|_2^2\right)\right)$$

$$= -\left[-\frac{d_x}{2}\log(2\pi\sigma_0^2) - \frac{1}{2\sigma_0^2}\|w^{(k)} - \theta \cdot \mathbf{1}_{d_x}\|_2^2\right]$$

$$= \frac{d_x}{2}\log(2\pi\sigma_0^2) + \frac{1}{2\sigma_0^2}\|w^{(k)} - \theta \cdot \mathbf{1}_{d_x}\|_2^2.$$

Since the first term does not depend on either $w^{(k)}$ or $\theta$, it can be considered an additive constant for $U$, and is thus absorbed into the constant $c$. Indeed, the relevant part of this negative log-prior is

$$\frac{1}{2\sigma_0^2}\|w^{(k)} - \theta \cdot \mathbf{1}_{d_x}\|_2^2.$$

Lastly, combining the relevant terms, ignoring the constant $c$, we arrive at

$$U(w^{(k)}, \theta|\mathcal{D}_{\text{train, val}}) = \sum_{i=1}^{|\mathcal{D}_{\text{train, val}}|} \left[\log(1 + e^{x_i^\top w^{(k)}}) - y_i x_i^\top w^{(k)}\right] + \frac{1}{2\sigma_0^2}\|w^{(k)} - \theta \cdot \mathbf{1}_{d_x}\|_2^2,$$

as desired.

### D.1.2   Derivation of the Hessian upper bound

Here, we outline the derivation of $H_{\text{bound}}$, the weight-independent upper bound for the Hessian matrix of the single-particle energy function $U(w, \theta|\mathcal{D}_{\text{train,val}})$ with respect to the regression weights $w$. To be clear, the purpose of $H_{\text{bound}}$ is to estimate the global Lipschitz constant $L$ of the gradient $\nabla_w U$. This constant $L$, taken as the largest eigenvalue of $H_{\text{bound}}$, subsequently informs the calculation of a baseline step-size $h_{\text{euler}}$ used in constructing grids for tuning the particle update step-size.

Recall, as seen above, single-particle energy function is given by:

$$U(w, \theta | \mathcal{D}_{\text{train, val}}) = \underbrace{\sum_{i=1}^{|\mathcal{D}_{\text{train, val}}|} \left[ \log \left( 1 + e^{x_i^\top w} \right) - y_i x_i^\top w \right]}_{\text{Negative Log-Likelihood (NLL)}} + \underbrace{\frac{1}{2\sigma_0^2} \| w - \theta \cdot \mathbf{1}_{d_x} \|_2^2}_{\text{Negative Log-Prior (NLP)}}.$$

We are then interested in the Hessian matrix $\nabla_w^2 U(w, \theta | \mathcal{D}_{\text{train}})$, which we decompose as

$$\nabla_w^2 U = \nabla_w^2(\text{NLL}) + \nabla_w^2(\text{NLP}).$$

Since $\text{NLP}(w, \theta) = \frac{1}{2\sigma_0^2} \| w - \theta \cdot \mathbf{1}_{d_x} \|_2^2$, we observe the gradient, with respect to $w$, to be $\nabla_w \text{NLP}(w, \theta) = \frac{1}{\sigma_0^2}(w - \theta \cdot \mathbf{1}_{d_x})$, and so this means that the Hessian of the NLP term, with respect to $w$, is thus constant

$$\nabla_w^2(\text{NLP}) = \frac{1}{\sigma_0^2} I_{d_x}.$$

The NLL term, on the other hand, gives a Hessian of the form

$$\nabla_w^2(\text{NLL}) = \sum_{i=1}^{|\mathcal{D}_{\text{train,val}}|} x_i \sigma(x_i^\top w)(1 - \sigma(x_i^\top w)) x_i^\top,$$

which can notably be expressed in matrix form as $X_{\text{train,val}}^T D(w) X_{\text{train,val}}$, where $X_{\text{train,val}}$ is a $|\mathcal{D}_{\text{train,val}}| \times d_x$ matrix, and $D(w)$ is a $|\mathcal{D}_{\text{train,val}}| \times |\mathcal{D}_{\text{train,val}}|$ diagonal matrix, with its $i$-th diagonal element being $d_{ii}(w) = \sigma(x_i^\top w)(1 - \sigma(x_i^\top w))$. Crucially, $\sigma(z)(1 - \sigma(z))$ is bounded, and is specifically maximised at $z = 0$, where $\sigma(0) = 0.5$, yielding a maximum value of $0.5 \times (1 - 0.5) = 0.25$.

Therefore, for all $x_i$ and $w$,

$$0 < \sigma(x_i^\top w)(1 - \sigma(x_i^\top w)) \leq 0.25,$$

implying that each (diagonal) element $d_{ii}(w) \leq 0.25$, so that the matrix $D(w) \preceq 0.25 I_{|\mathcal{D}_{\text{train,val}}|}$ (in the *Loewner order*).

As a result, an upper bound for the NLL Hessian is given by

$$\nabla_w^2(\text{NLL}) = X_{\text{train,val}}^T D(w) X_{\text{train,val}} \preceq 0.25 X_{\text{train,val}}^T I_{|\mathcal{D}_{\text{train,val}}|} X_{\text{train,val}} = 0.25 X_{\text{train,val}}^T X_{\text{train,val}}.$$

Combining the Hessian for the NLP term with the upper bound for the NLL Hessian, we notably obtain a weight-independent upper bound for the total Hessian $\nabla_w^2 U(w, \theta | \mathcal{D}_{\text{train,val}})$,

$$\nabla_w^2 U \preceq \frac{1}{4} X_{\text{train,val}}^T X_{\text{train,val}} + \frac{1}{\sigma_0^2} I_{d_x},$$

and so define the constant matrix $H_{\text{bound}}$ as this quantity. To be clear, we have

$$H_{\text{bound}} = \frac{1}{4} X_{\text{train,val}}^T X_{\text{train,val}} + \frac{1}{\sigma_0^2} I_{d_x}. \tag{37}$$

The largest eigenvalue of this positive semi-definite matrix $H_{\text{bound}}$ provides the global Lipschitz constant $L$ used to determine $h_{\text{euler}}$.

### D.2 Error model selection - Bayesian regression

### D.2.1 Derivation of the covariance matrix for Gaussian marginal likelihood

For $\mathcal{M}_G$, we have the Bayesian linear regression model $y = Xw + \varepsilon$, with $w \sim \mathcal{N}(0, \alpha_0^{-1} I_{d_x})$, $\varepsilon \sim \mathcal{N}(0, \sigma_0^2 I_{d_y})$, and $w$ and $\varepsilon$ independent.

Now, the mean of $y$ is $E[y] = E[Xw + \varepsilon] = XE[w] + E[\varepsilon] = 0$, while the covariance matrix $\Sigma_0 = \text{Cov}(y)$ is given by

$$\Sigma_0 = E[(y - E[y])(y - E[y])^\top] = E[yy^\top].$$

Expanding $yy^\top$ gives

$$yy^\top = (Xw + \varepsilon)(Xw + \varepsilon)^\top = Xww^\top X^\top + Xw\varepsilon^\top + \varepsilon w^\top X^\top + \varepsilon\varepsilon^\top,$$

which gives, upon taking the expectation, and using $E[w\varepsilon^\top] = E[w]E[\varepsilon^\top] = 0$, due to independence and zero means,

$$\Sigma_0 = E[Xww^\top X^\top] + E[\varepsilon\varepsilon^\top],$$
$$= XE[ww^\top]X^\top + E[\varepsilon\varepsilon^\top].$$

Since $E[ww^\top] = \text{Cov}(w) + E[w]E[w]^\top = \alpha_0^{-1}I_{d_x}$ and $E[\varepsilon\varepsilon^\top] = \text{Cov}(\varepsilon) + E[\varepsilon]E[\varepsilon]^\top = \sigma_0^2 I_{d_y}$, we find

$$\Sigma_0 = \sigma_0^2 I_{d_y} + \alpha_0^{-1}XX^\top,$$

where $\Sigma_0$ is the covariance of $y$ under the model, $p(y|X, \sigma_0^2, \alpha_0) = \mathcal{N}(y|0, \Sigma_0)$, as desired.

### D.2.2   ML-II estimation for $\mathcal{M}_G$

Type-II Maximum Likelihood (ML-II ) estimation determines hyperparameters by maximising the marginal likelihood. Specifically, we achieve this by minimising the negative log-marginal likelihood with respect to the logarithmic parameterisations, $\phi_1 = \log \sigma^2$ and $\phi_2 = \log \alpha$.

The log marginal likelihood for $\mathcal{M}_G$ is

$$L(\phi_1, \phi_2) = \log p(y|X, \sigma^2 = e^{\phi_1}, \alpha = e^{\phi_2}),$$
$$= -\frac{d_y}{2}\log(2\pi) - \frac{1}{2}\log(\det(\Sigma)) - \frac{1}{2}y^\top\Sigma^{-1}y,$$

where $\Sigma = e^{\phi_1}I_{d_y} + e^{-\phi_2}XX^\top$.

The ML-II estimates for $(\phi_1, \phi_2)$ are found by solving $\arg\min_{\phi_1, \phi_2} -L(\phi_1, \phi_2)$ This minimisation is performed numerically, typically using gradient-based optimisation algorithms such as L-BFGS-B, which we choose to utilise.

