# OpenReview forum: "Learning Latent Variable Models via Jarzynski-adjusted Langevin Algorithm"
_NeurIPS.cc/2025/Conference — NeurIPS 2025 poster_

### Official Review · Reviewer_iWiS · 2025-06-09

**Clarity:** 4
**Significance:** 3
**Originality:** 4
**Rating:** 5
**Confidence:** 5

**Summary:**

In this paper, the authors introduce the Jarzynski-adjusted Langevin algorithm (JALA) as a sampling procedure for sequences of evolving probability measures. JALA can be viewed as a type of Expectation–Maximisation algorithm to perform maximum marginal likelihood estimation (MMLE) in latent variable models (LVMs).  Building on nonequilibrium statistical mechanics, JALA runs biased Unadjusted Langevin Algorithm (ULA) dynamics on a time‐varying potential and applies an exponential Jarzynski weight as a bias correction to the estimates of expectations under the target measures. This sampler is then embedded in an EM‐style routine—termed JALA-EM—in which a particle population evolves under ULA, weights are updated via Jarzynski factors, and parameter updates are performed using a generic first‐order optimiser, e.g. SGD, although other optimisers could be used.

The authors provide a nonasymptotic convergence analysis of JALA-EM under log‐concavity and Polyak–Łojasiewicz conditions, showing that with a suitable choice of step‐size and particle count the method converges to the MMLE with an explicit rate of the form
$\E\bigl[V(\theta_k)-V^\star\bigr] \le \delta_0 (1-\tfrac{\gamma\mu}{2})^k + O\bigl(\tfrac{\gamma}{N}\bigr) + O\bigl(\tfrac{1}{N}\bigr),$
where $V(\theta)=-\log p_\theta(y)$ and $N$ is the number of particles.

Empirically, JALA-EM is evaluated on (i) Bayesian logistic regression on the Wisconsin breast cancer dataset, demonstrating faster, robust convergence of both parameter estimates and posterior approximations compared to particle gradient descent (PGD) and SOUL; (ii) Bayesian linear regression model selection, where JALA-EM yields accurate marginal‐likelihood estimates on‐the‐fly and correctly identifies the data‐generating model; and (iii) a small Bayesian neural network on MNIST binary digit classification.

**Questions:**

1) Have the authors tested JALA-EM against SOUL, PGD or other related methods on higher-dimensional examples? And if so, how does JALA-EM compare?
2) The authors do not comment on the sensitivity of tuning parameters such as $gamma$ and $h$. The theoretical results do not seem to provide a clear indication of how these parameters should be tuned in practice. What advice should be given to practitioners regarding step size selection so that JALA-EM can be easily used out-of-the-box?
3)

**Ethical Concerns:**

["NO or VERY MINOR ethics concerns only"]

**Final Justification:**

There is a clear consensus among the reviewers to accept the paper. I'm happy with the quality of the paper and its appropriateness for the NeurIPS conference. The authors also addressed the concerns/questions that I raised.

**Limitations:**

Yes

**Paper Formatting Concerns:**

No issues

**Quality:**

4

**Strengths And Weaknesses:**

Quality

The paper’s technical quality is high, underpinned by a rigorous nonasymptotic convergence analysis of the JALA–EM algorithm under clearly defined, and reasonable, assumptions such as smoothness and the Polyak–Łojasiewicz condition. This theoretical treatment, uncommon in hybrid sequential Monte Carlo–EM work, provides practitioners with explicit error bounds and convergence rates, lending strong credibility to the method. Empirically, the authors validate their claims through comprehensive benchmarks—including Bayesian logistic regression, linear regression model selection, and a small neural network on MNIST—comparing against state-of-the-art baselines (PGD and SOUL) with well-tuned hyperparameters and wall-clock timing. A small weakness is that the analysis hinges on strong assumptions (e.g., bounded gradients and finite exponential moments of Jarzynski weights) that may not hold in high-dimensional or strongly multimodal settings, potentially limiting the method’s practical applicability. However, the authors do note that some assumptions could be relaxed, albeit leading to a more complex theoretical analysis. Moreover, the experimental scale remains quite modest—datasets of dimension up to nine or small Bayesian neural networks—so it remains unclear how JALA-EM will perform on large, complex latent-variable models. It could also be reasonably argued that the empirical study could be improved if the authors compared JALA-EM against other competing EM-type methods, i.e. not just SOUL and PGD.

Clarity

Overall, the paper is well structured: the algorithmic description (Algorithm 1), theoretical propositions, and implementation details are presented in discrete, logical sections that make the paper clear to follow. Notation is consistent with standard conventions in both SMC and Langevin literature, which makes the paper easy to read for those familiar with this literature. That said, the work assumes significant prior familiarity with nonequilibrium thermodynamics (the Jarzynski equality) and advanced SMC formalisms, which could present a steep learning curve for those not already versed in both domains. For example, in equation (2), it isn't immediately clear why $A_k$ is needed (unless the reader is familiar with the Jarzynski method, which is not well-known in the sampling literature), whereas the ULA process on $X_k$ is clear as this is the latent variable to be estimated.

Significance

A central strength of the proposed approach is its ability to recursively estimate marginal likelihoods “on the fly” during parameter learning, something most scalable EM-type and particle-gradient methods do not offer. This feature enables principled model selection within the same algorithmic run that fits parameters, representing a meaningful advance for practitioners who require both posterior inference and evidence computation. Furthermore, by introducing tools from nonequilibrium statistical mechanics into mainstream latent-variable inference, the paper bridges disciplines which should open new avenues of research between these two areas. However, the necessity of maintaining and updating Jarzynski weights for each particle introduces extra computational and memory overhead compared to vanilla ULA or particle-gradient schemes, potentially negating runtime benefits in some applications. The reliance on resampling to control weight degeneracy also raises concerns, which are well-known in the SMC literature. Selecting effective ESS thresholds or weight-clipping strategies may lead to non-trivial tuning and could introduce additional variance in practice.

Originality

Methodologically, the use of Jarzynski–adjusted Langevin dynamics within an SMC ensemble to drive an EM algorithm for marginal‐likelihood estimation is genuinely novel and clearly extends prior SMC-EM and Jarzynski-based sampling work. The theoretical contribution—providing the first nonasymptotic convergence guarantees for an SMC–EM algorithm leveraging nonequilibrium corrections—is particularly noteworthy. Nonetheless, the innovation builds fairly directly on recent work by Carbone et al. (2024a,b) and existing SMC-EM variants, so some readers may view the advance as incremental rather than revolutionary. Additionally, the introduction of new hyperparameters (e.g., the Jarzynski step-size and ESS resampling threshold) adds further complexity to the tuning process, which could make the algorithm more challenging for practitioners to successfully implement in practice.

---

> ### Author Rebuttal · Authors · 2025-07-30
>
> We are thankful to the reviewer for their thoughtful and informative review.
>
> > it remains unclear how JALA-EM will perform on large, complex latent-variable models.
>
> To address the point about complex LVMs (as also asked by other reviewers), we have extended our example on classification with BNNs beyond the binary class setting, where a clear example is one in which we consider MNIST samples corresponding to more than two distinct digits. For example, we classify 2500 subsampled images corresponding to the digits 2, 4, 7, and 9 (rather than just 4 and 9), for a variety of BNN model capacities. To be clear, the original BNN has 40 hidden neurons, and thus for the 4-way classification task has 31,520 latent variables. For $N=50$, $K=500$, and again a common global step-size, we obtain average final misclassification error rates (and standard deviations) of 4.74% (0.34%), 9.24% (0.93%), and 9.20% (1.27%) for PGD, JALA-EM, and SOUL respectively. As alluded to, one can vary the number of hidden neurons, and to this end we repeated the experiment for 512 hidden neurons, resulting in a BNN of 403,456 latent variables, and average final misclassification error rates (and standard deviations) of 4.98% (0.42%), 7.20% (0.81%), and 8.02% (1.02%) for PGD, JALA-EM, and SOUL respectively.
>
> > It could also be reasonably argued that the empirical study could be improved if the authors compared JALA-EM against other competing EM-type methods, i.e. not just SOUL and PGD.
>
> Many thanks for this comment. We specifically focus on cases where E and M steps both require numerical algorithms (e.g. sampling for E-step and optimisation for M-step). We believe SOUL and PGD are the canonical choices in this setting and best performing benchmarks. This is because these methods leverage Langevin-type sampling ideas to implement the sampling and gradient-based optimisers for M-step. Other benchmarks (e.g. using Metropolis-Hastings for E step or gradient-free optimisation for M-step) would be interesting to consider but would underperform given the complexity of the problems we consider here. We will mention these approaches in our work.
>
> > The work assumes significant prior familiarity with nonequilibrium thermodynamics (the Jarzynski equality) and advanced SMC formalisms, which could present a steep learning curve for those not already versed in both domains. For example, in equation (2), it isn't immediately clear why $A_{k}$ is needed (unless the reader is familiar with the Jarzynski method, which is not well-known in the sampling literature), whereas the ULA process on $X_{k}$ is clear as this is the latent variable to be estimated.
> >
>
> Many thanks for this comment. We agree the method as written in Prop. 1 deviates from the standard way of presenting it in other sequential Monte Carlo papers (where $A_k$ (log-weights) are not directly mentioned, but weights are). We will provide the following remark in our paper to make a connection to more standard SMC literature:
>
> **Remark.** *We note that, in connection to more standard SMC literature, the sequence $(A_k)_{k\geq 0}$ denote *log-weights*. In other words, let us denote weights associated with our particles $\mathsf{w}_k^i$ for $i = 1,\ldots, N$. In our context, we denote $A_k = \log \mathsf{w}_k$ and $\mathsf{w}_k = e^{A_k}$. Hence it can be seen that the formula in Prop. 1 and in Algorithm 1 implements a weighted average.*
>
> > Have the authors tested JALA-EM against SOUL, PGD or other related methods on higher-dimensional examples? And if so, how does JALA-EM compare?
> >
>
> Thank you for mentioning this, as the performance of JALA-EM in higher-dimensional settings is important. First, note that the original BNN model used for the binary classification of 4 and 9 MNIST digits, in the paper, has $(28 \times 28 \times 40) + (40 \times 2) = 31,440$ latent weights. In the multi-class setting, this is expanded to $(28 \times 28 \times 40) +(40 \times 4) = 31,520$ latent weights, as mentioned above, whereas the larger model has $(28 \times 28 \times 512) +(512 \times 4) = 403,456$ latent weights, providing high-dimensional problem(s) as requested. As outlined above, JALA-EM is comparable to SOUL which is to be expected due to the correlations that exist between particles due to the resampling mechanism present in JALA-EM, and again note that PGD does not support marginal likelihood estimation.
>
>
> > The authors do not comment on the sensitivity of tuning parameters such as $\gamma$ and $h$. The theoretical results do not seem to provide a clear indication of how these parameters should be tuned in practice. What advice should be given to practitioners regarding step size selection so that JALA-EM can be easily used out-of-the-box?
> >
>
> Thank you for highlighting this critical point, since this is also a common challenge for the other particle algorithms considered. To this end, as done in Appendix B.2 of our paper (see the **Approach.** section), we suggest utilising a systematic and reproducible procedure for model fitting and hyperparameter selection. In particular, we perform K-fold cross-validation on $\mathcal{D}\_{train, val}$ to select step-sizes. In our case, we have utilised a pre-defined grid for the particle step-sizes, and where applicable (i.e. for JALA-EM and SOUL) for the $\theta$-update step-sizes also. To form the grid of step-sizes, relevant to the particle updates, we compute a baseline step-size, which we denote $h\_{euler}$, and the idea here is to provide a theoretically grounded scale reflecting the curvature of the energy landscape. More details pertaining to these grids can notably be found in both Appendix B.2. and Appendix C.1.2. The evaluation metric utilised for this hyperparameter tuning can of course be chosen at the discretion of the practitioner, however we opted for the Log Pointwise Predictive Density (LPPD), which assesses the model’s average predictive accuracy on unseen data. To be clear, the step-size combination yielding the highest average LPPD across the folds is chosen.
>
> We hope that this cross-validation scheme serves as an initial systematic way in which step-sizes can be chosen, ensuring our method can be used *out-of-the-box*. We agree that further refinement of this process would be a promising direction for future research.
>
> We are happy to discuss any of these points further if the reviewer has any more concerns.

---

> > ### Comment · Reviewer_iWiS · 2025-08-05
> >
> > Thank you to the authors for your response to my questions. I have also read the responses that you have given to the other reviews and I am very pleased with the thoroughness of your responses. I think this is a great paper, which was reflected by my initial score, and I will maintain my score (which also seems to align with the other reviewers).

---

### Official Review · Reviewer_QYFn · 2025-07-02

**Clarity:** 3
**Significance:** 3
**Originality:** 4
**Rating:** 5
**Confidence:** 3

**Summary:**

The authors use jarzynksi-adjusted langevin sampling from nonequilibrium statistical mechanics to develop a weight-version of sequential monte carlo methods.

**Questions:**

How does the algorithm compare to algorithms such as Hamiltonian sampling for sampling speed? What is the difference in performance between using the Jarzynski sampling for time-varying probability distributions and sequential Hamiltonian sampling as in Septier and Peters, IEEE, 2015?

**Ethical Concerns:**

["NO or VERY MINOR ethics concerns only"]

**Final Justification:**

The authors clearly addressed my concerns.  I appreciate the comparison to OLS and BIC in the rebuttal.  While it is still unclear how this algorithm performs in comparison to momentum-based approaches, I believe that is outside the scope of this manuscript and it, in and of itself, is a solid foundation for work in the future.

**Limitations:**

Well addressed in the paper.

**Paper Formatting Concerns:**

As a minor edit, the references are distracting being a different color and seem like they take up a lot of space.

**Quality:**

3

**Strengths And Weaknesses:**

Strengths - The authors describe many ways in which the algorithm may be used in neural networks.  It is flexible and metrics are well established to describe how sampling distribution converges to marginal likelihood.  I thank the authors for their clear derivations and writing. I think this paper will be of interest to those who use Bayesian neural networks.

Weaknesses - The sampling speed is not clear to be. Generally, Kl Divergence over time is used to compare speed. As authors mentioned, it is computationally expensive due to the weights, so it is not clearly written when this method would be better than gradient descent that makes long jumps such as levy flights.

---

> ### Author Rebuttal · Authors · 2025-07-30
>
> We thank the reviewer for their thoughtful and informative review.
>
> > Weaknesses: The sampling speed is not clear to be. Generally, Kl Divergence over time is used to compare speed. As authors mentioned, it is computationally expensive due to the weights, so it is not clearly written when this method would be better than gradient descent that makes long jumps such as levy flights.
>
> We appreciate the reviewer’s observation. It is true that, in our method, computing and applying the importance weights introduces a computational overhead compared to existing algorithms, such as PGD and SOUL. However, we emphasize that the final iteration complexity of our method is the same as these two methods If we denote the cost of a single evaluation of the gradient of $U$ by $C_{\nabla U}$, then we have that the  computational (iteration) cost of PGD, SOUL, and JALA is $\mathcal{O}(N C_{\nabla U})$. We appreciate this obsfucates practical speeds of the respective methods, and considerations related to parallelisation. As indicated by Figure 1 in our paper, our method is practically slower than PGD, due to multiple potential and gradient evaluations, and less parallelizable compared to PGD, due to the synchronization bottleneck introuced by the resampling step.
>
> In any case, we note that our method enables recursive estimation of the marginal likelihood, hence its applicability to model selection, which recall the PGD method cannot facilitate.
>
> With respect to gradient descent, we would like to note that our method is an instance of SGD when OPT is chosen as SGD. JALA-EM can make long jumps, depending on the variance of the gradient estimator. However, we remark that this can be beneficial or harmful depending on the model under consideration.
>
> > How does the algorithm compare to algorithms such as Hamiltonian sampling for sampling speed
>
> We thank the reviewer for the suggestion. In this work, we did not include a benchmark against momentum-based samplers such as Hamiltonian Monte Carlo or underdamped Langevin dynamics, since our method is built on top of ULA-type dynamics, which are overdamped and do not involve auxiliary momentum variables. The theoretical framework can, in principle, be extended to underdamped schemes, and we agree that such an extension would be a promising direction for future research.
>
>
> > What is the difference in performance between using the Jarzynski sampling for time-varying probability distributions and sequential Hamiltonian sampling as in Septier and Peters, IEEE, 2015?
> >
>
> Thank you for suggesting this reference. [Septier & Peters, IEEE, 2015] consider a filtering setting (as opposed to our setting where we are interested in parameter estimation in a static model). Therefore, it is not directly possible to provide a comparison between two approaches. That being said, the Hamiltonian proposals in their work can also be adapted to our setting. While they focused on Metropolis-adjusted proposals, our focus was on the Jarzynski-based correction framework to unadjusted overdamped dynamics, and in particular its use in the (static) latent variable estimation context. We will cite this work in our literature review.
>
> [Septier & Peters, IEEE, 2015] Septier, F., & Peters, G. W. (2015). Langevin and Hamiltonian based sequential MCMC for efficient Bayesian filtering in high-dimensional spaces. IEEE Journal of selected topics in signal processing, 10(2), 312-327.
>
> > Paper Formatting Concerns: As a minor edit, the references are distracting being a different color and seem like they take up a lot of space.
> >
>
> We thank the reviewer for the observation. In the updated version, we will switch to another color to improve readability to reduce visual clutter.
>
> We are happy to discuss or clarify if the reviewer has any more questions.

---

> > ### Comment · Reviewer_QYFn · 2025-08-04
> >
> > Thank you for the clarifications and discussions! I have no further questions and will raise my score.

---

### Official Review · Reviewer_j23q · 2025-07-03

**Clarity:** 3
**Significance:** 3
**Originality:** 3
**Rating:** 5
**Confidence:** 4

**Summary:**

This paper makes use of recent advances in the application of the Jarzynski relation and sequential monte carlo to estimate expectations under target distributions with densities known up to normalization. Here, it is used to perform an estimation of an expectation necessary in an expectation-maximization algorithm for latent variable models, e.g. to perform maximum marginal likelihood estimation.

The authors make use of the Jarzynski relation to sample under a potential $U_k$ as specified in a previous work [1] by keeping track of importance weights necessary to adapt a local unadjusted langevin dynamics (ULA). The target expectation here is one emerging in the optimization of latent variable models, i.e. to find the maximizers of the marginal likelihood $p_\theta (y)$ of the joint model $p_\theta(x,y)$.

They test their proposed method on bayesian logistic regression and bayesian neural network tasks and show that it is robust and comparable to existing approaches while offering the ability to simultaneously estimate a number of quantities.









[1] Carbone et al, "Efficient Training of Energy-Based Models Using Jarzynski Equality", 2023.

**Questions:**

- In the notation, I may be missing something. Can the authors clarify why the dynamics in 6 gives access to the expectation (through the use of the weights) of the *conditional density p(x|y)*? Or are you just using Bayes rule and not saying it somewhere? I ask because $U(\theta, x)$ is defined as $- \log p_\theta(x,y)$ and not $-\log p_\theta(x|y)$.

- Can the authors more clearly delineate why their approach gives access to certain quantities that the alternative approaches do not? This is mentioned throughout, but it is not clear to me why all the other methods are not amenable to any sort of online likelihood estimation / partition function estimation. Is this truly not available in other approaches? This would help me understand the contribution here a bit better. It does not need be some unique miracle to be a valuable contribution, I just want to clarify that fulfilling a variety of these desiderata in one method is not amenable to the alternative approaches.

- Could you provide a few more experimental realizations? This is really lacking in terms of differentiating with other approaches.

- Could you comment on the sensitivity of the proposed method to the resampling procedure and the frequency of resampling?

**Ethical Concerns:**

["NO or VERY MINOR ethics concerns only"]

**Final Justification:**

The clarification of new experimentation and the novelty of the asymptotic analysis is useful. I recommend an accept.

**Limitations:**

yes

**Quality:**

3

**Strengths And Weaknesses:**

*Strengths*

- The paper does a good job introducing the method, with mostly clear writing, and provides a nice theoretical analysis of the implications of how the proposed algorithm must be implemented (e.g. how to control biases, etc.)
- Numerical experiments validate the robustness of the approach, and show that it is competitive with existing approaches, while also providing potentially more estimable quantities simultaneously (I want the authors to comment more about this distinguishing factor).


*Weaknesses*

- Methodologically, the jarzynski formulation was already completely typified as it is used here in [1]. I guess the authors see the key insight as reapplying it in a new domain, i.e. to estimate a different expectation on which to perform gradient descent. This is a bit lacking.
- One could argue that applying it in a new domain (which definitely requires new insights!) is sufficient to warrant support of the work, but it is hard to tell of the value of the method given the limited numerical experiments as it pertains to existing approaches. The only comparisons done to other methods is on the Bayesian neural network example, and the authors write that the performance of JALA-EM is comparable to that of SOUL, and slower than PGD. The authors do note that their method provides greater access to certain quantities of interest, like an online estimate of the partition function, but they should make clearer why this is unique to their approach (perhaps with a table of characteristics of the various methods).




[1] Carbone et al, "Efficient Training of Energy-Based Models Using Jarzynski Equality", 2023.

---

> ### Author Rebuttal · Authors · 2025-07-30
>
> We thank the reviewer for their thoughtful and informative review.
>
> > Weaknesses: Methodologically, the jarzynski formulation was already completely typified as it is used here in [1]. I guess the authors see the key insight as reapplying it in a new domain, i.e. to estimate a different expectation on which to perform gradient descent. This is a bit lacking.
>
> We thank the reviewer for pointing this out. We agree that the formulation we employ builds upon existing work, notably [1]. Our methodological contributions are two-fold:
>
> - Adapting the methodology in [1] to a statistical problem (MMLE) as pointed out by the reviewer - expanding its scope to a whole class of statistical models.
> - Providing a convergence analysis for this scheme, connecting the literature on biased gradient descent and IS-type estimates of gradients of marginal likelihoods.
>
> This second aspect is also crucial to our contribution as the theoretical analysis of $\theta$ iterates was left open in [1]. We will emphasise this a bit more in our updated manuscript.
>
> > Weaknesses: One could argue that applying it in a new domain (which definitely requires new insights!) is sufficient to warrant support of the work, but it is hard to tell of the value of the method given the limited numerical experiments as it pertains to existing approaches. The only comparisons done to other methods is on the Bayesian neural network example, and the authors write that the performance of JALA-EM is comparable to that of SOUL, and slower than PGD. The authors do note that their method provides greater access to certain quantities of interest, like an online estimate of the partition function, but they should make clearer why this is unique to their approach (perhaps with a table of characteristics of the various methods).
> >
>
> We thank the reviewer for these suggestions. First, we'd like to note further experiments are conducted (see our response below). To answer the last request, we have prepared a table to summarize when/how JALA-EM provides unique properties.
>
> |Feature|PGD|SOUL|JALA-EM|
> |-|:-:|:-:|:-:|
> |Marginal likelihood estimation|✖|✖|✔|
> |Different $\theta$ and $x$ step-sizes| ✖| ✔| ✔|
> |Theoretical guarantees with fixed $\gamma$| ✔| ✖|✔|
> Easy adaptation of theory to alternative optimizers|✖| ✖|✔|
>
> We will add this table to our paper. To expand the last point, we would like to emphasize another nontrivial aspect of our approach, that is, *easy adaptation of theory to alternative optimizers*. In practice, methods like PGD and SOUL are implemented using *adaptive optimizers*, e.g., see [2]. However, their theory can only account for vanilla gradient descent (for SOUL, theory only works for a particular (and non-trivial) decay of step-sizes). In our case, due to the SMC-based approach to gradient estimation, we can apply theory for *any optimizer* provided that guarantees for biased gradients exist. We can easily also do this with adaptive optimizers, adapting results from, e.g., [3]. We will expand these points in our updated manuscript.
>
> > In the notation, I may be missing something. Can the authors clarify why the dynamics in 6 gives access to the expectation (through the use of the weights) of the conditional density p(x|y)? Or are you just using Bayes rule and not saying it somewhere? I ask because $U(\theta, x)$ is defined as $-\log p_{\theta}(x, y)$ and not $-\log p_{\theta}(x \vert y)$.
> >
>
> We are indeed implicitly using the standard fact from sampling theory that the posterior $p_\theta(x \mid y)$ is proportional to the joint density $p_\theta(x, y)$, that is $-\log p_\theta(x \mid y) = -\log p_\theta(x, y) + \log p_\theta(y)$, where the marginal $\log p_\theta(y)$ is independent of $x$ and thus irrelevant for the dynamics or the importance weights. Therefore, using the joint log-density $-\log p_\theta(x, y)$ as the energy function in the ULA dynamics is equivalent to targeting the unnormalized posterior. This is a standard approach in Langevin-based sampling, where only the gradient of the log-density is required up to an additive constant. We will make this point explicit in the revised version to avoid confusion.
>
> > Can the authors more clearly delineate why their approach gives access to certain quantities that the alternative approaches do not? This is mentioned throughout, but it is not clear to me why all the other methods are not amenable to any sort of online likelihood estimation / partition function estimation. Is this truly not available in other approaches? This would help me understand the contribution here a bit better.
>
> In general, Markov chain Monte Carlo (MCMC) algorithms cannot readily provide estimates of marginal likelihoods (normalising constants). This is the main difficulty behind the approaches like SOUL to provide estimates of these quantities. There are, however, ways to provide estimates of the normalizing constant using MCMC output, see, e.g. [4]. However, these classical approaches are post-processing techniques -- means that unlike JALA-EM, they would require the whole chain for estimation -- rather than providing estimates online.
>
> PGD [2] approach simulates an SDE to minimise an energy functional and similarly they do not provide a way to estimate normalising constants. In fact, it is not clear how one can use their samples to obtain this estimate, unlike the MCMC case above.
>
> > Could you provide a few more experimental realizations? This is really lacking in terms of differentiating with other approaches.
>
> Thank you for the suggestion - in an effort to alleviate your concern, we have extended the BNN example to a multi-class setting, where specifically we extended the MNIST binary classification task described to a multi-class task, using 2500 subsampled images corresponding to the digits 2, 4, 7, and 9, for a variety of BNN model capacities. The original BNN has 40 hidden neurons, and thus 31,520 latent weights. For $N=50$, $K=500$, and again a common global step-size, we obtain average final misclassification error rates (and stds) of 4.74% (0.34%), 9.24% (0.93%), and 9.20% (1.27%) for PGD, JALA-EM, and SOUL respectively. Indeed, one can vary the number of hidden neurons, and so we repeated the experiment for 512 hidden neurons, resulting in a BNN of 403,456 latent weights, and average final misclassification error rates (and stds) of 4.98% (0.42%), 7.20% (0.81%), and 8.02% (1.02%) for PGD, JALA-EM, and SOUL respectively.
>
> For an additional Bayesian model selection example, we implemented a polynomial regression example. Rather than comparing two models, $\mathcal{M}\_{G}$ and $\mathcal{M}\_{T}$, we consider a series of nested models, $\mathcal{M}\_{1}, \dots,\mathcal{M}\_{10}$, where $\mathcal{M}\_{p}$ denotes a polynomial of order $p$. We then use estimated marginal likelihoods to select the one best supported by the evidence.
>
> To robustly evaluate performance we iterate through a range of true orders $p\_{true} \in [2,8]$, and run $100$ trials with different random seeds for each order. For each trial, we randomly sample $\boldsymbol{w}\_{true} \in \mathbb{R}^{p\_{true}+1}$, which is then multiplied by the respective expanded polynomial basis matrix $X$, for $M=500$ datapoints, and then perturbed by random Gaussian noise ($\sigma=7.5$). For each noisy dataset, the task is to select the best model from a candidate set of polynomial degrees $p \in [1, 10]$.
>
> We set $K = 200$ and $N=50$, selecting the model with the highest final estimated marginal likelihood. The parameters estimated here are $\phi\_{1} = \log(\sigma^{2})$ and $\phi\_{2} = \log(\alpha)$, where $\alpha$ is the precision of a zero-mean Gaussian prior placed on the weights. As a baseline, we compare our method to an approach in which the model weights estimated via OLS and the selected model has the lowest Bayesian Information Criterion (BIC).
>
> Performance is evaluated using the Mean Absolute Error (MAE) to provide a nuanced measure, rewarding close estimates when not perfectly accurate, which is not interpretable from just the Accuracy, for example. The MAE across the 100 repeats is summarised below:
>
> |True Order|JALA-EM (MAE)|OLS & BIC (MAE)|
> |-|:-:|:-:|
> |2|0.41|0.35|
> |3|0.30|0.34|
> |4|0.33|0.30|
> |5|0.35|0.35|
> |6|0.24|0.23|
> |7|0.12|0.15|
> |8|0.16|0.20|
>
> Code for both of these new examples will be made available in the future in the public repository.
>
> > Could you comment on the sensitivity of the proposed method to the resampling procedure and the frequency of resampling?
> >
>
> Regarding the resampling procedure, we did not explore alternatives beyond the standard systematic resampling method already used in [1], which is known to offer a good trade-off between computational efficiency and variance control. As discussed in that work, systematic resampling tends to be stable across a range of settings, and we believe that further refinements in this direction would yield marginal benefits compared to other factors, such as controlling the variance of the importance weights, which plays a more critical role in the performance of the method.
>
> Concerning the resampling frequency, we adopted a conservative strategy by setting the threshold parameter $C = 1/1.05$ to ensure a high empirical ESS at each step. While this choice helps maintain particle diversity, we agree that a more systematic study of the sensitivity of the algorithm with respect to C could be a valuable follow-up.
>
> We are keen to discuss if there are any further questions.
>
> [1] Carbone, Davide, et al. "Efficient training of energy-based models using jarzynski equality." (2023), *NeurIPS*.
>
> [2] Kuntz, Juan, et al. "Particle algorithms for maximum likelihood training of latent variable models." (2023), *AISTATS*
>
> [3] Surendran, Sobihan, et al. "Non-asymptotic analysis of biased adaptive stochastic approximation." (2024), *NeurIPS*
>
> [4] Chib, S., & Jeliazkov, I. (2001). Marginal likelihood from the Metropolis–Hastings output. Journal of the American statistical association, 96(453), 270-281.

---

> > ### Comment · Reviewer_j23q · 2025-08-04
> > **Follow up**
> >
> > Thanks for your clarifications! I am increasing my score to 5.

---

### Official Review · Reviewer_un9b · 2025-07-03

**Clarity:** 3
**Significance:** 3
**Originality:** 3
**Rating:** 5
**Confidence:** 4

**Summary:**

A special case of Del Moral et al's SMC framework applied to Type 2 MLE optimization and similar problems, where the forward kernel is ULA, the backward kernel is the standard choice for deterministic involutions, and the sequence of distribution is given by the top level parameters optimization path.

**Questions:**

An in any Z estimator, we need to know the true value of Z_0, e.g. this is shown in (4). In traditional SMC Samplers, we choose pi_0 so that it can have a known normalization constant (e.g. the prior, or a variational distribution). However in the context of the paper, how is Z_0 estimated (especially in the high dimensional case, where simple IS would fail)? This is an important thing to clarify as the paper highlights "JALA-EM’s unique and dual capability to concurrently estimate both parameters and the marginal likelihood".

**Ethical Concerns:**

["NO or VERY MINOR ethics concerns only"]

**Final Justification:**

Thanks for the nice discussion! No major adjustment needed in my score.

**Limitations:**

Yes

**Paper Formatting Concerns:**

Problem in citation style: utilising the results from (Demidovich et al., 2023).

rather should take N large. -> rather, it is enough to take N large.

Similarly, M-step -> Similarly, the M-step

Missing caps in bib, e.g. Ascent-based monte carlo

**Quality:**

3

**Strengths And Weaknesses:**

A strength of the manuscript is the non-asymptotic convergence analysis. As the authors correctly point out, the fact that the gradient estimation is biased makes the analysis less routine than typical stochastic gradient theoretical analysis.

The paper is well written and explores interesting ideas. I enjoyed reading it. I hope you find these comments below helpful...

As my above summary description emphasizes, the paper is firmly within the Del Moral 2006 / Neal 2001 framework of SMC Samplers/AIS with backward kernels, something that the authors do explain in remark 1. As a result I feel introducing a new terminology for the methodology "Jarzynski-adjusted Langevin algorithm" is just going to add more noise in the literature. In fact, that combination of unadjusted moves with AIS/SMC Sampler keeps being re-invented and renamed. One example of re-invention in the ML literature is "Markov Chain Monte Carlo and Variational Inference: Bridging the Gap" by Salimans, Kingma, Welling 2015 (another great example of sketchy citation practice: that 2015 paper cites other works of Neal but not a word of the AIS paper it is re-inventing...).

Just to be clear, I find that combination of AIS/SMC with unadjusted kernels applied to Type 2 MLE interesting, and it becomes more novel when it is combined with the optimization of top level parameters. But I strongly feel the new name is a disservice to the community! An easy change, but one I would feel strongly about.

Here is a specific example why it would be better to rely more on the original literature: let us look at (4) in Proposition 1. The estimator for the normalization constant shown there corresponds to the AIS case, i.e. no resampling. Yet the authors do use resampling later on (as they should!), but in 5.2 they appear to still use the AIS estimator. If instead of relying on a re-invention of the wheel (here, citing Carbone et al 2024), they used the original, i.e. (15) in Del Moral 2006, they would obtain a much improved estimator.

Another point that I think should be changed is to present the algorithm as taking OPT to be a black box algorithm. Here the problem is that if OPT goes too far in the parameter space, the distributions pi_k and pi_{k+1} will be very far from each other, which in turn leads to weight degeneracy.   The correct formalism would be to take OPT to be a trust-region optimizer. This is related to the Adaptive SMC Samplers literature where an annealing parameter is tuned adaptively (see e.g. Zhou Johansen Aston 2015). Here the fact that a single step of SGD creates an implicit notion of "not going too far in one step".

"as exact maximisation is intractable" -> that's a bit imprecise... often while the overall objective is non-convex, the inner M-step can still be convex. Often when an optimizer within an M step is used, it is because the optimizer does not have close form solution.

In practice, adaptive step sizes are important for large scale deployment of SMC samplers, here in two ways. First, in terms of controlling the distance between the successive distribution (this is related to my earlier comment on trust region methods), it is typical that a non-constant step size is needed to avoid degeneracy (see again Zhou 2015). Second, more specifically to this manuscript, there is the step size in the state space x. For the latter, a good reference in the context of unadjusted algorithms with SMC/AIS sampler is Kim, Xu, Gardner, Campbell, 2025 and references therein.

The experiments are interesting but jump straight from toy models to a black box NN example. I wish there would have been something in between, e.g. a spatial model or a scientific application.

One obvious numerical comparison that I would have found interesting is to compare the unadjusted SMC with the adjusted SMC one (i.e., the latter being the traditional choice where the backward kernel is the time reversal), while keeping as much as possible the same (except for the step size which would need to be different for the two).

---

> ### Author Rebuttal · Authors · 2025-07-30
>
> We thank the reviewer un9b for their thoughtful and informative review.
>
> **Name of the method/algorithm**: We would like to address first the concern of the reviewer saying (we do not copy paste it all due to the length limit):
>
> > I strongly feel the new name is a disservice to the community! An easy change, but one I would feel strongly about.
>
> We fully agree with the reviewer's message about not adding noise in terms of inventing new algorithm names. We also agree with the reviewer's assessment here that our method is a typical SMC algorithm with unadjusted Langevin kernels/proposals. In this respect, one possible 'name' for our method could be "SMC with ULA proposals". This would however become quickly untenable in naming also our parameter estimation method.
>
> Our perspective adopting our acronym is to take a dual perspective on the method. While we fully agree the reviewer's perspective on seeing the method as "SMC with ULA proposals", we take the MCMC perspective here on seeing it as an "adjustment" of ULA - much like Metropolis adjustment. This adjustment takes the form of running identical chains and correcting them using weights, based on Jarzynski identity. Given the roots of the ideas in the statistical physics literature, and also recognising Jarzynski's contributions, we named this special adjusted ULA variant as JALA. Our intention is definitely not to add the noise - but to come up with a specific name to this variant - which has been not specifically named in the literature.
>
> Finally, to explain our perspective, we would like to give an example from MCMC literature. For example, Metropolis-Hastings is a general method. However, a special instance of this method (MH with Langevin proposals) is termed MALA in the literature. We took here a similar approach, we hope that the reviewer sees this as a recognition of this special subclass of methods. We have been also very careful in Section 2 to summarize the relations to Del Moral (2006). We are open to discussion in the rebuttal phase on this point. We will definitely also add the citation [1] the reviewer requested in another comment as this seems very relevant.
>
> > [...] let us look at (4) in Proposition 1. The estimator for the normalization constant shown there corresponds to the AIS case, i.e. no resampling. Yet the authors do use resampling later on (as they should!), but in 5.2 they appear to still use the AIS estimator. If instead of relying on a re-invention of the wheel (here, citing Carbone et al 2024), they used the original, i.e. (15) in Del Moral 2006, they would obtain a much improved estimator.
> >
>
> Apologies for any misunderstanding here. We will indeed fix the formula in our Section 5.2, as we believe this is already identical to original formula as pointed out by the reviewer in the implementation. That is, after resampling, we reset the weights as in Del Moral (2006).
>
> > Another point that I think should be changed is to present the algorithm as taking OPT to be a black box algorithm. Here the problem is that if OPT goes too far in the parameter space, the distributions pi_k and pi_{k+1} will be very far from each other, which in turn leads to weight degeneracy. The correct formalism would be to take OPT to be a trust-region optimizer. This is related to the Adaptive SMC Samplers literature where an annealing parameter is tuned adaptively (see e.g. Zhou Johansen Aston 2015). Here the fact that a single step of SGD creates an implicit notion of "not going too far in one step".
>
> Thank you and we fully agree with this excellent observation as we have seen this in practice. Our optimisers need to take 'small enough' steps for distributions to stay close and remain easy to sample. We will write a remark on this point on the choice of OPT and its impact on sampling performance and will definitely mention the necessity of adaptive methods and will cite the relevant references in our remark.
>
> > "as exact maximisation is intractable" -> that's a bit imprecise...
>
> We agree with this comment. We have reworded this part, now reads:
>
> *as exact maximisation may not be analytically available.*
>
> > *Need for adaptive step sizes*: In practice, adaptive step sizes are important for large scale deployment of SMC samplers, here in two ways. First, in terms of controlling the distance between the successive distribution (this is related to my earlier comment on trust region methods), it is typical that a non-constant step size is needed to avoid degeneracy (see again Zhou 2015). Second, more specifically to this manuscript, there is the step size in the state space x. For the latter, a good reference in the context of unadjusted algorithms with SMC/AIS sampler is Kim, Xu, Gardner, Campbell, 2025 and references therein.
> >
>
> We agree with this comment, and as we noted above, we will add a discussion on the need of adaptive step-sizes in our setting.
>
> > The experiments are interesting but jump straight from toy models to a black box NN example. I wish there would have been something in between, e.g. a spatial model or a scientific application.
>
> Thank you for this feedback - to bridge this gap we have implemented a new experiment, namely that of Bayesian model selection in a polynomial regression setting. The idea is that this serves as an intermediate step in complexity between the initial logistic regression example and the BNN example. Rather than comparing two models, $\mathcal{M}\_{G}$ and $\mathcal{M}\_{T}$, we consider a series of nested models, $\mathcal{M}\_{1}, \dots,\mathcal{M}\_{10}$, where $\mathcal{M}\_{p}$ denotes a polynomial of order $p$. We then use estimated marginal likelihoods to select the one best supported by the evidence.
>
> To robustly evaluate performance we iterate through a range of true orders $p\_{true} \in [2,8]$, and run $100$ trials with different random seeds for each order. For each trial, we randomly sample $\boldsymbol{w}\_{true} \in \mathbb{R}^{p\_{true}+1}$, which is then multiplied by the respective expanded polynomial basis matrix $X$, for $M=500$ datapoints (that we randomly sample in $[-2.5, 2.5]$), and is then subsequently perturbed by random Gaussian noise, determined by $\sigma=7.5$. For each generated (noisy) dataset, the task for the algorithm is to select the best model from a candidate set of polynomial degrees $p \in [1, 10]$.
>
> For our algorithm, we set $K = 200$ and $N=50$, selecting the model with the highest final estimated marginal likelohood. Note that the parameters estimated here are $\phi\_{1} = \log(\sigma^{2})$ and $\phi\_{2} = \log(\alpha)$, where $\alpha$ is the precision of a zero-mean Gaussian prior placed on the regression weights, and we initialise both values at 0. As a strong baseline, we compare our method to an approach in which the model weights estimated via OLS and the selected model has the lowest Bayesian Information Criterion (BIC).
>
> Performance is evaluated using the Mean Absolute Error (MAE) to provide a nuanced measure, rewarding close estimates when not perfectly accurate. The MAE across the 100 repeats is summarised below:
>
> |True Order|JALA-EM (MAE)|BIC & OLS (MAE)|
> |-|:-:|:-:|
> |2|0.41|0.35|
> |3|0.30|0.34|
> |4|0.33|0.30|
> |5|0.35|0.35|
> |6|0.24|0.23|
> |7|0.12|0.15|
> |8|0.16|0.20|
>
> ​The code for this new example will be made available in the future in the public repository.
>
> > One obvious numerical comparison that I would have found interesting is to compare the unadjusted SMC with the adjusted SMC one (i.e., the latter being the traditional choice where the backward kernel is the time reversal), while keeping as much as possible the same (except for the step size which would need to be different for the two).
>
> We agree that this is indeed a very interesting direction. There are multiple ways to implement these methods, e.g. a few options are summarised in [1]. While we could not provide this comparison now, we think that (with small enough step-sizes) these methods would perform parameter estimation similarly. We reference an interesting work that supports this view where some alternative approaches are characterised and assessed in the context of MCMC methods [2].
>
> > An in any Z estimator, we need to know the true value of Z_0, e.g. this is shown in (4). In traditional SMC Samplers, we choose pi_0 so that it can have a known normalization constant (e.g. the prior, or a variational distribution). However in the context of the paper, how is Z_0 estimated (especially in the high dimensional case, where simple IS would fail)? This is an important thing to clarify as the paper highlights "JALA-EM’s unique and dual capability to concurrently estimate both parameters and the marginal likelihood".
> >
>
> Indeed, the estimation of $Z\_0$ is a crucial aspect in any Z-estimator framework. In our setting, where $\pi\_0$ is not explicitly normalized, we therefore used importance sampling to estimate $Z\_0$. However we agree that for higher-dimensional models, this presents a genuine challenge where simple importance sampling becomes unreliable. One option to solve this issue is to apply a dedicated SMC/AIS procedure to bridge the initial posterior distribution with a tractable reference distribution and estimate its normalising constant. This can easily be integrated then into our approach and be updated.
>
> [1] Kim, K., Xu, Z., Gardner, J. R., & Campbell, T. Tuning Sequential Monte Carlo Samplers via Greedy Incremental Divergence Minimization. In Forty-second International Conference on Machine Learning (ICML) 2025.
>
> [2] Schönle, Christoph, et al. "Sampling metastable systems using collective variables and Jarzynski–Crooks paths." Journal of Computational Physics, 2025.

---

> > ### Comment · Reviewer_un9b · 2025-08-04
> >
> > Thanks for the comments. I still disagree that a new name should be created (especially since resampling is used as pointed out in the rebuttal). Otherwise I appreciate the additional references, experiment, and commitment to release code.

---

### Official Review · Reviewer_Ygyo · 2025-07-04

**Clarity:** 4
**Significance:** 3
**Originality:** 3
**Rating:** 5
**Confidence:** 4

**Summary:**

This paper contributes to the literature of parameter inference in latent variable models using their marginal likelihood function. The key challenge in such parameter estimation problems is the inability to analytically obtain the marginal likelihood function by integrating out the latent variables. Correspondingly, much effort has been devoted to finding efficient ways to estimate the gradients of the marginal likelihood function so that parameter estimation can be done. This paper proposes a novel methodology for doing so using a combination of unadjusted Langevin algorithm steps to drive a set of particles, and, crucially, compute a set of weights, that are used to estimate the gradient together with a bias correction and allow its use in conjunction with standard optimization algorithms. The authors provide some non-asymptotic results on the performance of method, giving some quantitative bounds on how quickly the sequence of iterates approaches the true value for a given sample size. Some - now standard for this literature - simple experiments are considered to demonstrate the efficacy of the methodology. Of interest in Bayesian statistics is estimation of the marginal likelihood function itself, and the proposed method is able to do that as well.

**Questions:**

Is Carbone's correction the only one that can be used naturally with ULA like this? Are there any other possible bias correction schemes for ULA? Does it rely on a particular discretization scheme?

Given that these are particle-based estimates of marginal likelihoods, what are their variance properties like?

I am very interested in unpacking Remark 1 a bit more here... how does the backward kernel connect to bias correction? Can you say a little more about this point please.

**Ethical Concerns:**

["NO or VERY MINOR ethics concerns only"]

**Final Justification:**

All of my questions have been satisfactorily clarified. Thank you to the authors!

**Limitations:**

Yes.

**Paper Formatting Concerns:**

None.

**Quality:**

3

**Strengths And Weaknesses:**

Strengths: the algorithm is certainly elegant and demonstrates how ULA samples can be directly bias-corrected. It is easy to implement (in fact, I have personally tried to do so by adjusting some existing logistic regression code and succeeded on my part with a little work). The paper provides a strong and practical answer to a question which is very natural for this literature. The writing is very clear, and the intuition behind the method is supported by theoretical results.

Weaknesses: The Bayesian neural network example is not very convincing and feels overly simplified. It would have been nice to see the method pushed a bit further in terms of what it can do in practice. For example, to do it on a non-binary classification task. And to report the misclassification rate for the task. Also in the main paper, the authors should be more clear in describing what exactly the latent variables are in the logistic regression model (this is said in the Appendix, but to a novice reader this may not be obvious).

---

> ### Author Rebuttal · Authors · 2025-07-30
>
> > Weaknesses: The Bayesian neural network example is not very convincing and feels overly simplified. It would have been nice to see the method pushed a bit further in terms of what it can do in practice. For example, to do it on a non-binary classification task. And to report the misclassification rate for the task.
>
> We agree that a non-binary classification task would be interesting and have thus extended the MNIST binary classification task described to a multi-class task, where specifically we classify 2500 subsampled images corresponding to the digits 2, 4, 7, and 9 (rather than just 4 and 9), for a variety of BNN model capacities. To be clear, the original BNN has 40 hidden neurons, and thus for the 4-way classification task has 31,520 latent weights. For $N=50$, $K=500$, and again a common global step-size, we obtain average final misclassification error rates (and standard deviations) of 4.74% (0.34%), 9.24% (0.93%), and 9.20% (1.27%) for PGD, JALA-EM, and SOUL respectively. As alluded to, one can vary the number of hidden neurons, and to this end we repeated the experiment for 512 hidden neurons, resulting in a BNN of 403,456 latent weights, and average final misclassification error rates (and standard deviations) of 4.98% (0.42%), 7.20% (0.81%), and 8.02% (1.02%) for PGD, JALA-EM, and SOUL respectively.
>
> Additionally, we have implemented another Bayesian model selection example, specifically for polynomial regression. Rather than comparing two models, $\mathcal{M}\_{G}$ and $\mathcal{M}\_{T}$, we consider a series of nested models, $\mathcal{M}\_{1}, \dots,\mathcal{M}\_{10}$, where $\mathcal{M}\_{p}$ denotes a polynomial of order $p$. We then use estimated marginal likelihoods to select the one best supported by the evidence.
>
> To robustly evaluate performance we iterate through a range of true orders $p\_{true} \in [2,8]$, and run $100$ trials with different random seeds for each order. For each trial, we randomly sample $\mathbf{w}\_{true} \in \mathbb{R}^{p\_{true}+1}$, which is then multiplied by the respective expanded polynomial basis matrix $X$, for $M=500$ datapoints (that we randomly sample in $[-2.5, 2.5]$), and is then subsequently perturbed by random Gaussian noise, determined by $\sigma=7.5$. For each generated (noisy) dataset, the task for the algorithm is to select the best model from a candidate set of polynomial degrees $p \in [1, 10]$.
>
> For our algorithm, we set $K = 200$ and $N=50$, selecting the model with the highest final estimated marginal likelohood. Note that the parameters estimated here are $\phi\_{1} = \log(\sigma^{2})$ and $\phi\_{2} = \log(\alpha)$, where $\alpha$ is the precision of a zero-mean Gaussian prior placed on the regression weights, and we initialise both values at 0. As a baseline, we compare our method to an approach in which the model weights are estimated via OLS and the selected model has the lowest Bayesian Information Criterion (BIC).
>
> Performance is evaluated using the Mean Absolute Error (MAE) to provide a nuanced measure, rewarding close estimates when not perfectly accurate, which is not interpretable from just the Accuracy alone, for example. The MAE across the 100 repeats is summarised below:
>
> |True Order|JALA-EM (MAE)|OLS & BIC (MAE)|
> |-|:-:|:-:|
> |2|0.41|0.35|
> |3|0.30|0.34|
> |4|0.33|0.30|
> |5|0.35|0.35|
> |6|0.24|0.23|
> |7|0.12|0.15|
> |8|0.16|0.20|
>
> ​The code for both of these new examples will be made available in the future in the public repository.
>
>
>
> >Also in the main paper, the authors should be more clear in describing what exactly the latent variables are in the logistic regression model (this is said in the Appendix, but to a novice reader this may not be obvious).
> >
>
> Many thanks for this suggestion - we have now clarified that the regression weights, $w \in \mathbb{R}^{d\_{x}=9}$, are the latent variables for this example. On reflection, although implicit through our Appendix, we agree it is better to be explicit about this in the main text.
>
>
> > Is Carbone's correction the only one that can be used naturally with ULA like this? Are there any other possible bias correction schemes for ULA? Does it rely on a particular discretization scheme?
> >
>
> Another example of such a correction is MALA. However, a key limitation is that particles evolve independently, without access to global information that could be used for resampling. As a result, when the energy landscape undergoes a phase transition—for example, from unimodal to bimodal—MALA may yield biased results if the proportion of particles in each mode does not match the correct distribution.
>
> The correction we propose is more general and can be applied to other Markov kernels, as illustrated in [1]. It does not depend on a specific discretization scheme. Instead, one simply needs to adapt the definition of the weights to the particular scheme chosen for the discretized SDE.
>
> > Given that these are particle-based estimates of marginal likelihoods, what are their variance properties like?
>
> We provide a discussion of the error bound in (10), which is generally nontrivial in the SMC setting. Regarding the variance, it is true that in general there are limited theoretical guarantees—a well-known challenge in the Sequential Monte Carlo literature. As we discussed in the text, variance can be controlled empirically through appropriate resampling strategies. While this control is typically based on the empirical Effective Sample Size (ESS) rather than the population ESS, this is precisely the framework in which practitioners operate in real applications.
>
> > I am very interested in unpacking Remark 1 a bit more here... how does the backward kernel connect to bias correction? Can you say a little more about this point please.
> >
>
> As discussed in Del Moral’s work, it is known that the variance of the importance weights is minimized when the backward kernel coincides with the true inverse of the forward kernel. However, in practice, such an inverse is rarely available in closed form. This has motivated a line of research aimed at learning or approximating the backward kernel to reduce variance, as explored for instance in [2, 3]. In our setting, we chose the backward kernel to match the forward dynamics, which is a standard and computationally efficient choice. However, we stress that this does not correspond to the inverse kernel, since the forward dynamics (e.g., ULA) is not symmetric and does not satisfy detailed balance. Consequently, the resulting importance weights are essential to correct for this asymmetry and ensure consistency. This point is central in recent contributions such as [1] we mentioned above, where it is shown that properly accounting for the lack of reversibility via the weights can significantly improve accuracy across a wide class of non-symmetric kernels. The correction applies independently of the choice of forward discretization scheme, as long as the associated weights are correctly computed. This flexibility is a strength of the method, allowing it to accommodate a variety of sampling kernels.
>
> [1] Carbone, D. (2025). Jarzynski Reweighting and Sampling Dynamics for Training Energy-Based Models: Theoretical Analysis of Different Transition Kernels. arXiv preprint arXiv:2506.07843.
>
> [2] Doucet, A., Moulines, E., & Thin, A. (2023). Differentiable samplers for deep latent variable models. Philosophical Transactions of the Royal Society A, 381(2247).
>
> [3] Kim, K., Xu, Z., Gardner, J. R., & Campbell, T. (2025) Tuning Sequential Monte Carlo Samplers via Greedy Incremental Divergence Minimization. In Forty-second International Conference on Machine Learning (ICML).

---

> > ### Comment · Reviewer_Ygyo · 2025-08-05
> >
> > Dear authors, thank you for these clear remarks and additional comments on your method. I appreciate the additional experiment as an improvement to the paper. I don't have any more questions on my end.

---

### Decision · Program_Chairs · 2025-09-17

**Decision:**

Accept (poster)

**Comment:**

# Summary

This paper introduces a novel method, referred to as the Jarzynski-Adjusted Langevin Algorithm, for maximum marginal likelihood estimation. The core challenge is estimating the gradient of the marginal likelihood which is an integral over the latent variables and is therefore intractable. JALA addresses this by running a SMC algorithm with an appropriate sequence of distributions and transitions given by the Unadjusted Langevin Algorithm applied to this sequence.

This approach yields a population of weighted particles, whose gradient estimates are used in a first-order optimization loop. The authors provide non-asymptotic convergence guarantees under standard conditions (log-concavity, Polyak–Łojasiewicz), showing explicit rates depending on particle count and step size.

Empirical results on Bayesian logistic regression, linear regression model selection, and a small Bayesian neural network show that JALA-EM offers competitive performance.

# Recommendation

All reviewers find the paper worth publication. and therefore I recommend acceptance.
I strongly encourage the authors to incorporate the reviewers’ feedback and points raised during the rebuttal discussion when preparing the final version of the paper.